# Fluent molecular mixing of Tau isoforms in Alzheimer's disease neurofibrillary tangles

Aurelio J. Dregni [1,3], Pu Duan[1,3], Hong Xu[2], Lakshmi Changolkar[2], Nadia El Mammeri[1], Virginia M.-Y. Lee[2] & Mei Hong [1✉]

Alzheimer's disease (AD) is defined by intracellular neurofibrillary tangles formed by the microtubule-associated protein tau and extracellular plaques formed by the β-amyloid peptide. AD tau tangles contain a mixture of tau isoforms with either four (4R) or three (3R) microtubule-binding repeats. Here we use solid-state NMR to determine how 4R and 3R tau isoforms mix at the molecular level in AD tau aggregates. By seeding differentially isotopically labeled 4R and 3R tau monomers with AD brain-derived tau, we measured intermolecular contacts of the two isoforms. The NMR data indicate that 4R and 3R tau are well mixed in the AD-tau seeded fibrils, with a 60:40 incorporation ratio of 4R to 3R tau and a small homotypic preference. The AD-tau templated 4R tau, 3R tau, and mixed 4R and 3R tau fibrils exhibit no structural differences in the rigid β-sheet core or the mobile domains. Therefore, 4R and 3R tau are fluently recruited into the pathological fold of AD tau aggregates, which may explain the predominance of AD among neurodegenerative disorders.

---

[1] Department of Chemistry, Massachusetts Institute of Technology, 170 Albany Street, Cambridge, MA 02139, USA. [2] Department of Pathology and Laboratory Medicine, Institute on Aging and Center for Neurodegenerative Disease Research, University of Pennsylvania School of Medicine, Philadelphia, PA 19104, USA. [3]These authors contributed equally: Aurelio J. Dregni, Pu Duan. ✉email: meihong@mit.edu

In the human brain, the microtubule-associated protein tau exists in six isoforms, which are mainly distinguished by the presence or absence of the second microtubule-binding repeat, R2 (ref. [1]). Tau isoforms that contain four microtubule-binding repeats (R1, R2, R3, and R4) are called four-repeat (4R) tau, whereas those that contain three microtubule-binding repeats (R1, R3, and R4) are called three-repeat (3R) tau. In a number of neurodegenerative diseases, these intrinsically disordered tau proteins become hyperphosphorylated and form intraneuronal aggregates. Alzheimer's disease (AD) is defined by these tau neurofibrillary tangles as well as the extracellular plaques formed by the β-amyloid peptide[2,3]. Interestingly, the AD tau tangles contain a mixture of 4R and 3R tau isoforms, whereas other tauopathies such as corticobasal degeneration and Pick's disease are characterized by either aggregated 4R tau or aggregated 3R tau. Recent cryo-electron microscopy (cryoEM) studies[4–6] revealed that there is a single well-defined molecular conformation for the β-sheet core of AD tau filaments, but did not determine how 4R tau and 3R tau are mixed in the aggregate. The rigid core of the AD tau filaments spans the R3, R4, and part of the R′ repeats, which are present in both 4R and 3R tau isoforms. Another mixed-tau disease, chronic traumatic encephalopathy, also exhibits a single rigid core structure for its tau filaments[7], with no discernible differentiation between 4R and 3R tau in the fibril. Understanding how different tau isoforms are mixed in these filaments is important, because extensive biochemical data have established that misfolded tau proteins in AD and other tauopathies spread by recruiting soluble tau protein into the aggregate, and distinct isoforms might be expected to interfere with the propagation of the amyloid structure[8–17]. To determine how 4R and 3R tau isoforms are mixed in the neurofibrillary tangles in AD brains, here we use solid-state nuclear magnetic resonance (NMR) spectroscopy to investigate AD-tau seeded [15]N-labeled and [13]C-labeled tau proteins. We show that the 4R and 3R tau isoforms are fluently mixed in AD tau filaments, with no detectable structural differences among pure 4R tau fibrils, pure 3R tau fibrils, and mixed 4R and 3R tau fibrils.

## Results

To determine the mode of mixing of 4R and 3R tau in AD tau aggregates, we seeded [15]N or [13]C-labeled recombinant tau monomers with 10% AD brain-derived sarkosyl-insoluble tau (AD tau) (Fig. 1a) and characterized the resulting fibrils using solid-state NMR. AD tau was pooled from ten neuropathologically confirmed diagnosis of AD (Supplementary Tables 1 and 2). The use of multiple AD brains ensures that the molecular structural results we obtain reflect the average properties of AD brain tau filaments instead of the property of one AD patient. We chose brains with short postmortem intervals and high tau-burdens at late stages of the disease[18,19]. AD tau seeds were sequentially extracted from these brains. To these AD tau extracts we added ninefold excess of either a 1:1 mixture of 0N4R and 0N3R tau monomers or one of the two monomers. After 7 days of incubation under shaking, about half of the added monomers became insoluble (Fig. 1b and Supplementary Fig. 1), indicating a fivefold increase in insoluble mass[18,19]. Unseeded recombinant tau remained largely soluble. Transmission electron microscopy (TEM) images reveal that the monomer-added samples formed much longer filaments (200–300 nm) than the original AD tau seeds (40–70 nm) (Fig. 2 and Supplementary Fig. 2a). Most of these filaments have a twisted morphology whereby an ~10 nm width alternates with an ~25 nm width. A small number of relatively straight fibrils with ~16-nm width is also observed (Supplementary Table 3). These two morphologies are similar to the paired helical filaments (PHF) and straight filaments (SF) in

AD tau[20], but differ from the long, straight and 14–23 nm wide fibrils assembled using heparin[21–23] (Fig. 2c and Supplementary Fig. 2b). The AD-tau templated fibrils can seed mouse tau in wild-type primary neurons into morphologically identical aggregates, as detected by immunofluorescence using a mouse-tau specific antibody (Fig. 1c). Recent studies showed that these AD-tau-seeded mouse tau aggregates are localized to dendrites and axons[19,24]. Quantification of the fluorescence intensities shows that the amount of aggregated mouse tau increased in the presence of templated fibrils compared to the 10% AD-tau control (Fig. 1d). Therefore, the templated recombinant tau fibrils are pathologically similar to AD tau; moreover, the templated fibrils have higher overall pathogenicity compared the original 10% AD tau seeds. A similar increase in pathogenicity were observed whether 4R tau alone, 3R tau alone, or a mixture of both isoforms were added to the AD tau seeds[25]. Together, these data indicate that the ultrastructural morphology and pathological activities of AD tau are faithfully propagated by recombinant tau, regardless of whether 4R tau, 3R tau or both isoforms were added.

To determine how 4R and 3R tau monomers are organized in AD-tau templated fibrils, we conducted a [15]N-[13]C rotational-echo-double-resonance (REDOR) NMR experiment[26] (Fig. 3). This experiment measures the proportion of [13]C-labeled tau that is one strand away from an [15]N-labeled tau in the hydrogen-bonded cross-β fibril. Because two hydrogen-bonded β-strand backbones are separated by ~4.8 Å, the distance-dependent intermolecular [15]N–[13]C dipolar coupling depends only on how two differentially labeled tau monomers are mixed (Fig. 3a, b). [13]C dipolar dephasing of an [15]N-labeled tau is manifested as intensity loss between a spectrum measured with the [13]C pulses off ($S_0$) and the [13]C pulses on ($S$). Greater [15]N–[13]C dipolar dephasing, or lower $S/S_0$ values, indicates a higher probability of [13]C-labeled tau following an [15]N-labeled tau. To increase the spectral sensitivity, we observed this [15]N intensity loss through the signals of the directly bonded amide proton. No [1]H spectral resolution is required, as the REDOR experiment detects all rigid amides in the rigid β-sheet fibril core (Fig. 3c, d). To simplify the [1]H spectra and facilitate the comparison of 2D [15]N–[1]H correlation spectra, we perdeuterated the protein monomers and back-exchanged them with water. The resulting recombinant tau contains only NH and OH protons.

We mixed 50% [15]N-labeled 4R tau and 50% [13]C-labeled 3R tau in two independently prepared AD-tau seeded samples ([AD] 4R[N]3R[C]) (Supplementary Table 4). A second pair of AD-tau seeded samples has the reverse mixed labeling motif, containing 50% [15]N-labeled 3R tau and 50% [13]C-labeled 4R tau ([AD] 3R[N]4R[C]). We first measured the REDOR dephasing ($S/S_0$) of heparin-fibrillized 0N4R tau samples[21,22] containing varying molar ratios of [15]N-labeled and [13]C-labeled monomers. These samples allow direct experimental calibration of the REDOR dephasing of different statistical mixtures of [15]N-labeled protein and [13]C-labeled protein. For these 0N4R tau samples, the probability of a [13]C-labeled monomer following an [15]N-labeled monomer ($p_{N→C}$) in the fibril is equal to the fraction of [13]C-labeled monomers in the mixture. The fact that in vitro-fibrilized tau and AD-seeded tau do not have the same rigid core structure does not affect the use of intermolecular REDOR for determining the statistics of mixing, because the REDOR experiment only depends on the existence of parallel-in-register cross-β packing, which is true for both heparin-fibrillized tau and AD tau.

Figure 3e shows the REDOR dephasing of mixed [13]C-labeled and [15]N-labeled 0N4R tau samples, whose [13]C:[15]N molar ratios range from 0:100% to 70%:30%. The dephasing curves show the expected trend: the higher the fraction of [13]C-labeled tau, the faster the dipolar dephasing, and the lower the $S/S_0$ values.

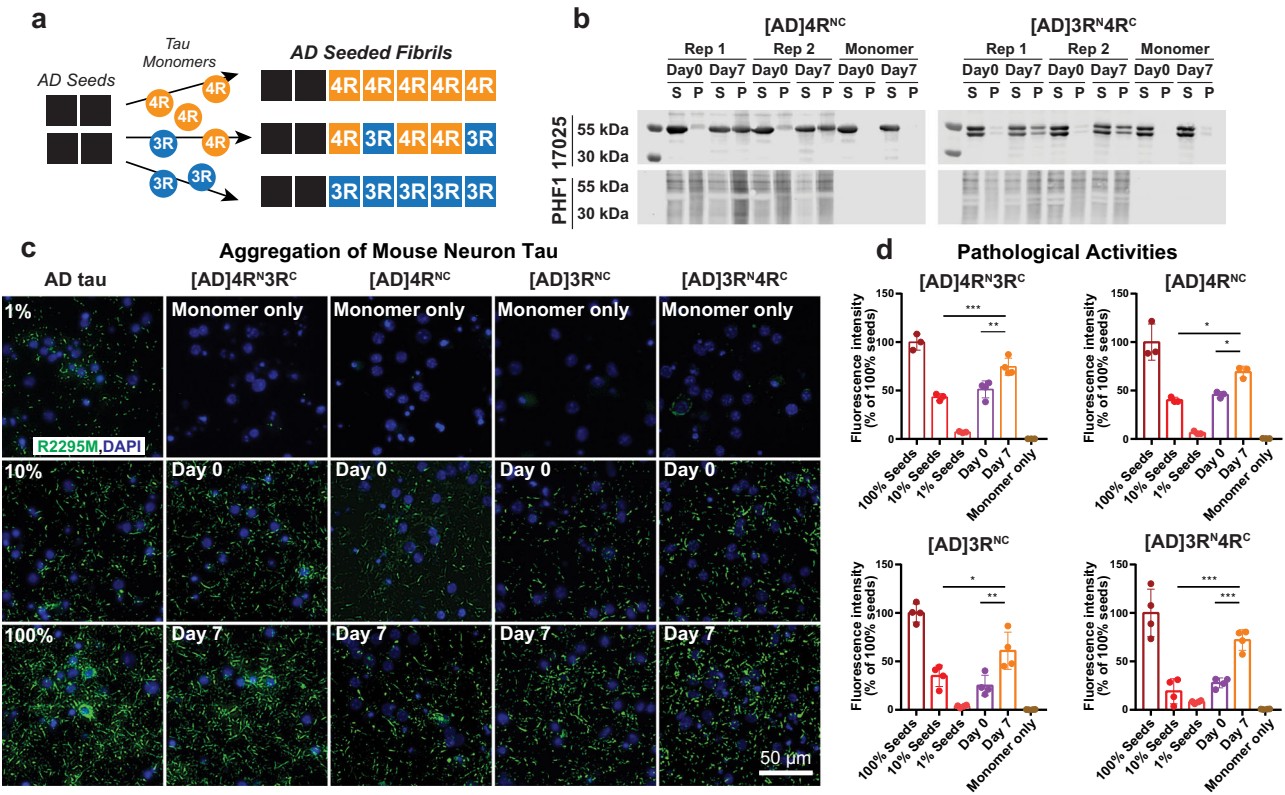

**Fig. 1 Isotopically labeled recombinant tau seeded with AD tau filaments conserves the AD pathogenic activity in wild-type neurons. a** Schematic of seeding recombinant 0N4R and 0N3R tau with AD tau extracts. **b** Sedimentation assay shows that recombinant tau monomers are incorporated into insoluble mass after 7 days of shaking with 10% AD tau seeds. Samples were fractionated into supernatant (S) and pellet (P) by ultracentrifugation. Tau species were revealed by the 17,025 antitotal tau antibody and PHF1 anti-phospho-tau antibody. Aliquots of replicate (Rep) samples 1 and 2 were collected for biochemical characterization at day 0 and day 7 (the end) of the fibrillization reaction. Reactions without AD tau seeds (monomer-only) were used as control to assess the extent of self-fibrillation. Full blots can be found in Supplementary Fig. 1. **c** Immunocytochemistry shows AD-tau-seeded recombinant tau fibrils recapitulate the pathogenicity of AD tau in primary neurons. Wild-type primary hippocampal neurons were transduced with AD tau or seeded tau fibrils. Aggregated tau pathology was revealed by R2295M anti-mouse tau antibody (green) and DAPI (blue) that stains cell nuclei. Seeded samples are AD-tau-seeded fibrils containing a 1:1 mixture of [15]N-labeled 4R tau and [13]C-labeled 3R tau, 4R-only tau, 3R-only tau, and a 1:1 mixture of [15]N-labeled 3R tau and [13]C-labeled 4R tau. **d** Quantification of fluorescence in **c** shows that the seeded fibrils at day 7 have increased pathogenicity compared to day 0. As expected, day 0 samples (10% AD tau seeds with monomers) showed similar pathogenicity levels to the 10% tau seeds without monomers. All values were normalized to the 100% AD tau seeds. Statistical analysis was performed using one-way ANOVA. Multiple comparison between groups was conducted using Tukey post hoc test. On the graph, *$P < 0.05$, **$P < 0.01$, ***$P < 0.001$ in the multiple comparison tests. Error bars are presented as standard deviation (SD). Each data point stands for one biological repeat: [AD]4R[N]3R[C] $n=3$, [AD]4R[NC] $n=3$, [AD]3R[NC] $n=4$, [AD]3R[N]4R[C] $n=4$.

The measured REDOR decay curves can be simulated using the known intermolecular distances in a cross-β amyloid fibril (Supplementary Fig. 3). To do this, we conducted both explicit spin simulations[27] and second-moment analysis[28]. Both methods of simulation reproduce the trend of the mixing-ratio-dependent dipolar dephasing, but do not match quantitatively the measured REDOR dephasing of the calibration samples. This is not surprising, because each simulation method has shortcomings that limit its accuracy (see "Methods"). The explicit spin simulation is restricted to a maximum of about 10 spins, while the second-moment analysis neglects nuclear spin interactions other than [15]N–[13]C dipolar coupling. Moreover, neither simulation captures the effects of natural abundance [13]C spins in the [15]N-labeled monomer. For these reasons, we rely on the more accurate experimental REDOR results of the calibration samples to extract the mixing probabilities of AD tau filaments.

The spectra of the [AD]4R[N]3R[C] samples yielded a REDOR dephasing curve that falls between the calibration curves for 30:70 and 40:60 [13]C:[15]N labeled 0N4R tau (Fig. 3e). Interpolation of the

measured $S/S_0$ values between the two calibration curves yielded a $p_{4 \rightarrow 3}$ value of $0.37 \pm 0.03$ (Supplementary Figs. 3 and 4). Thus, in AD-seeded tau filaments, a 4R tau is followed by a 3R tau with 37% probability, and hence by a 4R tau with 63% probability. The reversely mixed labeled [AD]3R[N]4R[C] samples showed faster REDOR dephasing: the measured REDOR curve falls between the calibration curves for 50:50 and 60:40 [13]C:[15]N 0N4R tau. Interpolation of the two measured calibration curves to match the REDOR dephasing of the AD-seeded sample yielded a $p_{3 \rightarrow 4}$ value of $0.56 \pm 0.02$. Thus, in the AD-seeded tau filaments, a 3R tau is followed by a 4R tau with 56% probability, and hence by a 3R tau with 44% probability. These data together indicate that each 4R tau monomer is flanked by 63% 4R tau, while each 3R tau monomer is flanked by 56% 4R tau. Therefore, the AD tau filaments, when offered equal amounts of 4R tau and 3R tau monomers, neither completely separate the two isoforms, which would correspond to $p_{4 \rightarrow 3} = p_{3 \rightarrow 4} = 0$, nor strictly alternate the two isoforms, which would correspond to $p_{4 \rightarrow 3} = p_{3 \rightarrow 4} = 1$. Moreover, both 4R tau and 3R tau have a small preference for contacting 4R tau. A two-state Markov model (see "Methods")

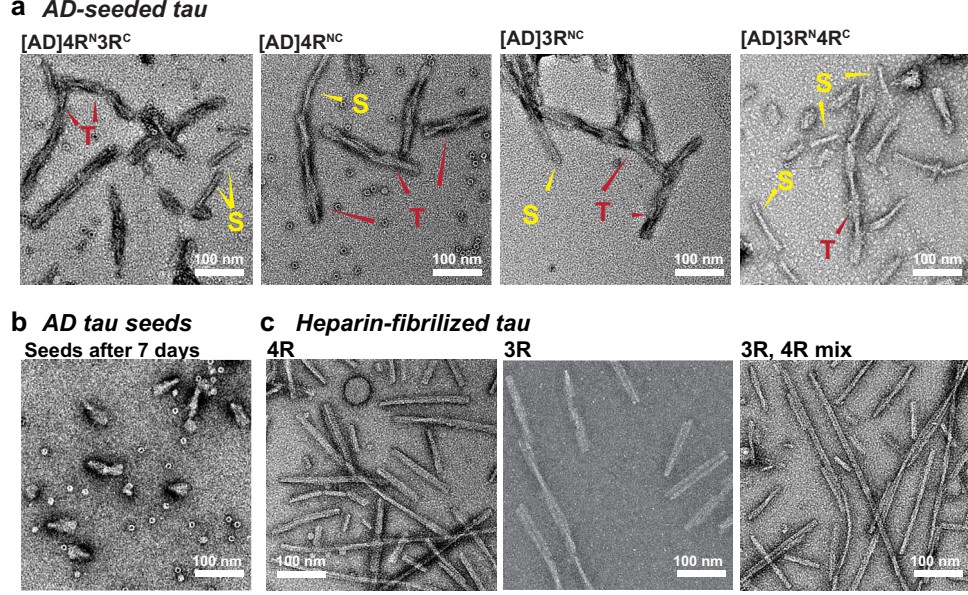

**Fig. 2 Ultrastructural morphology of AD-tau seeded tau fibrils is distinct from that of heparin-fibrillized tau. a** TEM images of AD-tau seeded fibrils containing a 1:1 mixture of $^{15}$N-labeled 4R tau and $^{13}$C-labeled 3R tau, 4R-only tau, 3R-only tau, and a 1:1 mixture of $^{15}$N-labeled 3R tau and $^{13}$C-labeled 4R tau. All samples show predominantly twisted (T) fibrils whose widths alternate between ~10 nm and ~25 nm, similar to the paired helical filaments of AD tau. In addition, a minority of straight (S) fibrils are observed, which are similar to the straight filaments of AD tau. **b** AD tau seeds after 7 days of incubation without added recombinant monomers. The seeds are fivefold shorter than the amplified tau fibrils. No long fibrils are observed. **c** Heparin-assembled recombinant tau fibrils. These fibrils are nearly exclusively straight, and are longer and more uniform than the straight filaments in AD-tau seeded filaments. At least five different images from different regions of each grid were obtained for each sample. A side-by-side comparison of these fibrils can be found in Supplementary Fig. 2.

shows that these probabilities can be converted to the mole fraction of 4R tau in the filament according to $\chi_4 = p_{3\rightarrow4}/(p_{3\rightarrow4} + p_{4\rightarrow3})$, giving a 4R tau mole fraction of $60 \pm 2\%$. Thus, the AD tau filaments incorporate the two isoforms with a 60: 40 molar ratio in favor of 4R tau. This composition is consistent with the estimated 4R and 3R tau incorporation levels from the $^{15}$N and $^{13}$C spectral intensities (Supplementary Fig. 5 and Supplementary Table 4). Furthermore, because $p_{4\rightarrow4}$ (63%) is larger than the overall 4R tau incorporation level (60%) while $p_{3\rightarrow4}$ (56%) is smaller than it, there is a small preference for homotypic 4R–4R and 3R–3R contacts over heterotypic 3R–4R contacts. We can alternatively express this preference for homotypic contact in terms of a mixing quotient, $Q = p_{3\rightarrow4}p_{4\rightarrow3}/p_{3\rightarrow3}p_{4\rightarrow4}$, defined in analogy to the equilibrium constant of mixing. Large $Q$ values indicate a preference for heterotypic mixing while small $Q$ values indicate a preference for block copolymerization. When there is no homotypic or heterotypic preference, i.e., when the mixing is agnostic of the present isoform, then $p_{3\rightarrow4} = p_{4\rightarrow4} = \chi_4$, and $Q = 1$. The measured $p_{4\rightarrow3}$ and $p_{3\rightarrow4}$ values correspond to a $Q$ value of $0.75 \pm 0.11$, indicating a small preference for homotypic contacts, independent of the overall isoform incorporation level. To provide a molecular view of this isoform mixing, we replicated the cryoEM structure of AD PHF tau[4] into a long filament containing 360 monomers in each protofilament, and simulated the 4R and 3R tau alternation to match the measured $p_{4\rightarrow3}$ and $p_{3\rightarrow4}$ values (Fig. 3f, g). The simulation depicts both the larger abundance of 4R tau over 3R tau in the filament and the intimate and fluent molecular mixing of the two tau isoforms.

To assess whether the AD-templated recombinant tau fibrils show isoform-dependent conformation and dynamics, we measured 2D $^{15}$N–$^1$H correlation NMR spectra. Dipolar hNH spectra selectively detect rigid β-sheet residues (Fig. 4a–e) whereas J-hNH INEPT spectra selectively detect nearly isotropically mobile

residues (Fig. 4f–i). In addition to the mixed isoform samples, we also prepared AD-tau seeded 4R-only tau and 3R-only tau samples. All four seeded tau fibrils show very similar dipolar hNH spectral patterns. For example, the chemical shifts of the resolved Gly peaks are the same. This AD-tau seeded spectral fingerprint is distinct from the peak pattern of heparin-fibrillized 4R tau. For example, the heparin-fibrillized sample shows resolved signals of the only two Cys residues in the protein and their sequential Gly residues (C291–G292 and C322–G323)[22]. These peaks are absent from the AD-tau seeded spectra. Therefore, the AD tau fold can recruit 3R tau monomers, 4R tau monomers, or a mixture of both, without changing the conformation of the rigid β-sheet core.

The 2D INEPT spectra (Fig. 4f–i) also show similar chemical shifts for the four AD-tau-seeded samples[22,23,29], indicating that the fuzzy coat has the same motionally averaged conformation for different tau isoforms. N-terminal residues to G207 at the beginning of the P2 domain exhibit the same chemical shifts and relative intensities. In comparison, C-terminal residues show weaker intensities and larger intensity variations, indicating that the C-terminus is less mobile compared to the N-terminus. In contrast to the dipolar spectra, the INEPT spectra of AD-tau-seeded samples are similar to the INEPT spectrum of heparin-fibrillized tau. Therefore, the mobile part of the fuzzy coat has similar dynamic conformations for all isoforms in both AD tau aggregates and in vitro fibrillized tau.

## Discussion
These data provide direct structural evidence of the nature of molecular mixing of 4R and 3R tau in AD brain tau filaments. The $^1$H-detected $^{15}$N-$^{13}$C REDOR experiment probes the molecular packing of tau isoforms along the fibril axis of AD tau, perpendicular to the C-shaped β-sheet structure[4]. We find that

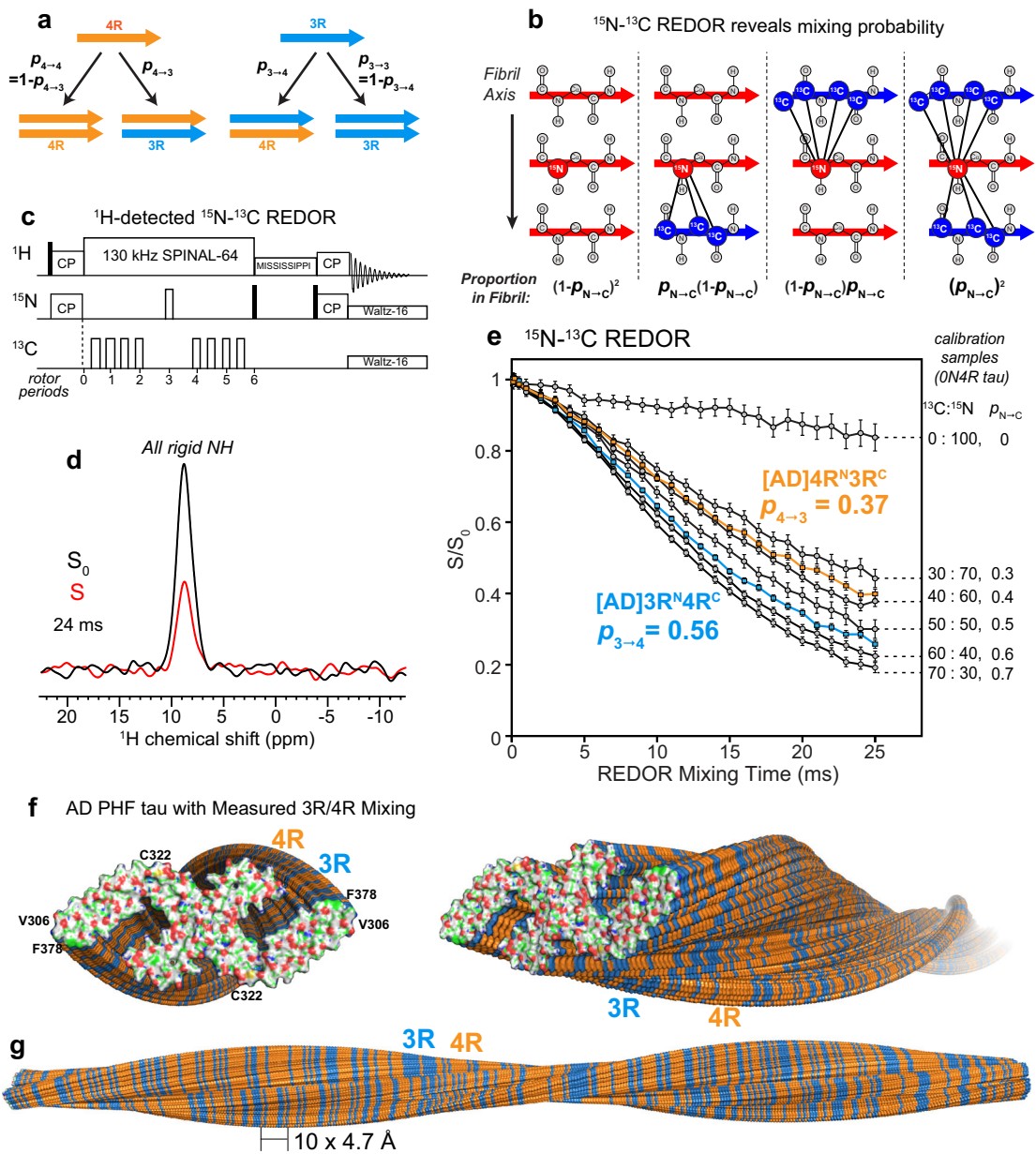

**Fig. 3 $^{15}$N-$^{13}$C REDOR NMR reveals the tau isoform mixing pattern in Alzheimer's disease brain tau filaments. a** Schematic of four possible isoform combinations of two hydrogen-bonded tau molecules and their probabilities. **b** Diagram of four possible nearest neighbors of an $^{15}$N-labeled tau and their probabilities in a fibril. The probability that an $^{15}$N-labeled tau is followed by a $^{13}$C-labeled tau is denoted by $p_{N \rightarrow C}$. Larger $p_{N \rightarrow C}$ values increase the intermolecular $^{15}$N-$^{13}$C dipolar dephasing, which is manifested as lower REDOR $S/S_0$ ratios. **c** Pulse sequence for the $^1$H-detected $^{15}$N-$^{13}$C REDOR experiment. **d** A representative pair of the REDOR spectra, showing the total intensities of all rigid protein amides in the absence ($S_0$) and presence ($S$) of $^{13}$C dephasing pulses. This spectrum was measured on [AD]4R$^N$3R$^C$ #2 with a mixing time of 24 ms under 20 kHz MAS. **e** Measured $^1$H-detected $^{15}$N-$^{13}$C REDOR decay curves. The REDOR data of mixtures of $^{13}$C, $^2$H-labeled 0N4R tau and $^{15}$N, $^2$H-labeled 0N4R tau serve as calibrations (gray) for quantifying $p_{N \rightarrow C}$, which is equal to the fraction of $^{13}$C-labeled protein in the mixture. The REDOR data of AD-seeded 4R$^N$3R$^C$ indicate a probability p$_{4 \rightarrow 3}$ of 0.37, while the data of AD-seeded 3R$^N$4R$^C$ give a probability p$_{3 \rightarrow 4}$ of 0.56. Each curve represents the average of two independently fibrillized samples. Error bars represent the propagated uncertainty from the spectral signal-to-noise ratio (one standard deviation). **f, g** Mixing of 4R (orange) and 3R (blue) tau in an AD PHF fibril (PDB: 5O3L). Tau isoforms along the fibril axis were simulated using the measured mixing probabilities. The simulations were done independently for each of the two protofilaments. **f** End view (left) and oblique view (right) of the PHF tau filaments. The terminal monomers' atoms are colored as H—white, C—green, N—blue, O—red, and S—yellow. **g** Sideview of the PHF tau fibril, showing a pair of protofilaments containing 360 monomers each, with a filament length of ~170 nm.

AD tau filaments recruit 4R and 3R tau monomers with a ratio of 60:40, and with a small preference for homotypic contacts. Moreover, the AD tau fibril incorporates 4R tau, 3R tau, or a mixture of the two isoforms without changing the β-sheet core structure, as seen by the similarity of the 2D dipolar hNH spectra (Fig. 4). Thus the AD tau prion incorporates either tau isoform into its pathological fold, without cross-seeding barrier between them, in contrast to in vitro assembled tau fibrils[30–32]. This fluent molecular mixing explains the biochemical observation that AD tau fibrils amplified with 4R tau, 3R tau, or both are equally capable of seeding both 4R and 3R mouse tau[18,19,25]. We hypothesize that this fluent mixing may promote fibril growth in vivo

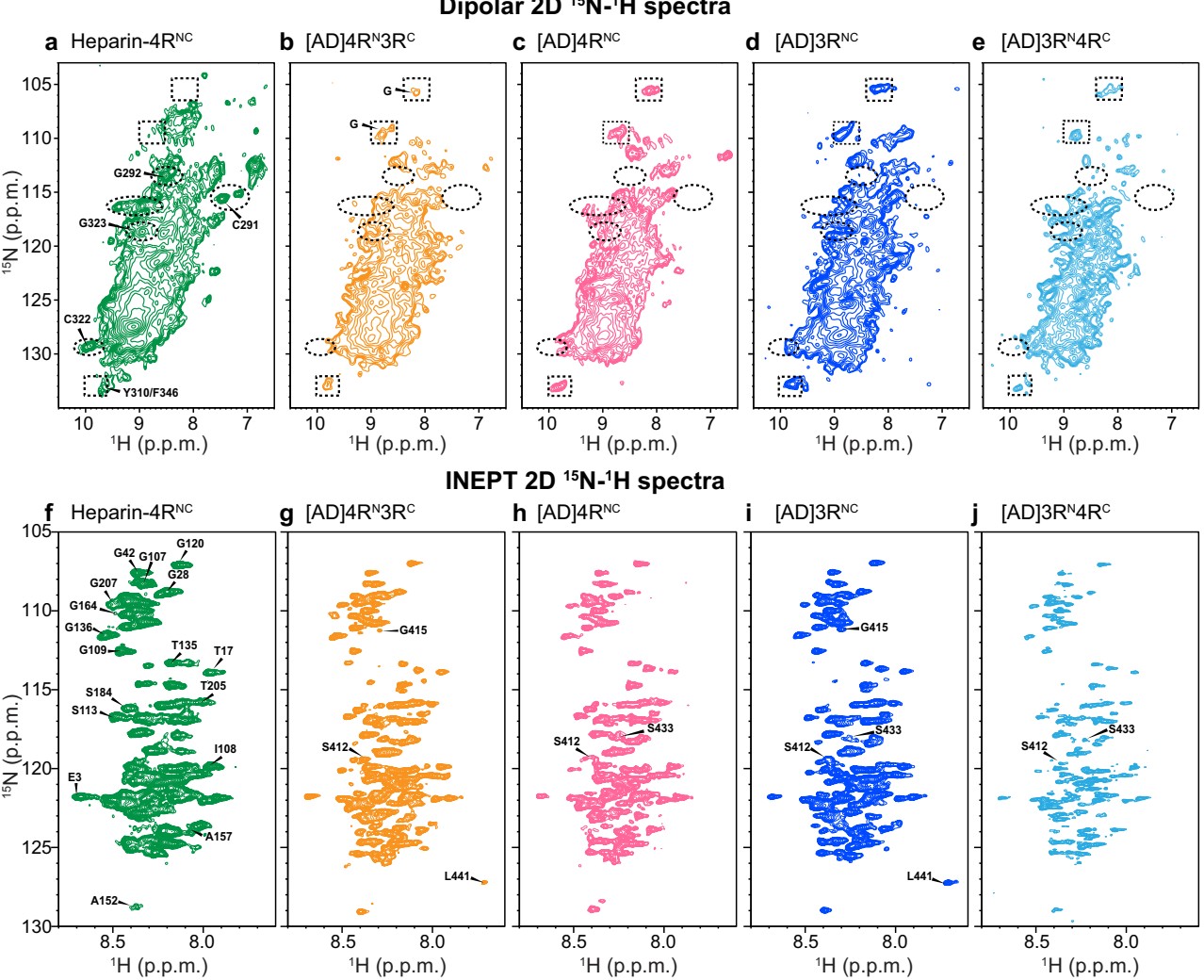

**Fig. 4 2D 15N-1H correlation spectra of AD-tau-seeded filaments show isoform-independent molecular conformation and dynamics that differ from heparin-fibrilized tau. a–e** 2D dipolar correlation spectra indicate a common rigid-core structure for all isoforms in AD seeded tau. **a** Heparin-fibrillized 0N4R tau. Several assignments based on 3D hCANH and hCA(co)NH spectra are indicated (Supplementary Fig. 7). **b** AD-tau seeded 15N-labeled 4R tau mixed with 13C-labeled 3R tau (1:1). **c** AD-tau seeded 15N, 13C-labeled 4R tau. **d** AD-tau seeded 15N, 13C-labeled 3R tau. **e** AD-tau seeded 15N-labeled 3R tau mixed with 13C-labeled 4R tau (1:1). Examples of peaks that are present in heparin-fibrillized tau but not in AD seeded tau are indicated by dashed ovals, while peaks that are present in AD seeded tau but not in heparin-fibrillized tau are indicated by dashed rectangles. **f–j** 2D INEPT spectra indicate similar dynamic segments for all isoforms in AD-seeded tau. Assignments for clearly resolved peaks are transferred from refs. [29,46]. **f** Heparin-fibrillized 0N4R tau. **g** AD-tau-seeded 15N-labeled 4R tau mixed with 13C-labeled 3R tau. **h** AD-tau-seeded 15N, 13C-labeled 4R tau. **i** AD-tau-seeded 15N, 13C-labeled 3R tau. **j** AD-tau seeded 15N-labeled 3R tau mixed with 13C-labeled 4R tau. Peaks from residues from the N-terminus to G207 are present in all samples, whereas C-terminal residues show varying intensities among the samples.

and play a role in making AD the most prevalent neurodegenerative disorder in humans.

Because the NMR data were obtained from AD tau seeds pooled from multiple patient brains, the 60:40 mixing and the probabilities we measured reflect the general properties of AD brain tau fibrils instead of the property of a single patient brain. The AD-seeded tau fibrils show highly reproducible 2D NMR spectra as well as reproducible REDOR dephasing, suggesting that there is no detectable molecular structural variation among the multiple AD brains. Although some biochemical studies have reported certain levels of heterogeneity in AD brain tau, these observations were all made based on total tau, including both pathogenic and non-pathogenic tau, in AD patients[33,34]. Our study instead focuses on sarkosyl-insoluble tau, which has been shown to have highly consistent fibril core structures across multiple AD patient cases[5], and which exhibits consistent seeding

abilities in vitro and in vivo[19]. Therefore, these prior biochemical and biophysical data provide additional support for the isoform-independent molecular structure observed here.

The 4R and 3R tau mixing probabilities we measured here are based on the assumption that the number of residues that contributes to the β-sheet core is the same for 4R and 3R tau in the AD-tau seeded fibrils. If the number of residues in the core differs between 4R and 3R tau, then the mixing probabilities would differ from the reported value. For example, if 4R tau incorporates more β-sheet residues into the fibril core than 3R tau, then the [AD] $4R^N3R^C$ REDOR $S/S_0$ intensities would be biased high, and the true $p_{4->3}$ value would be larger than the measured value. Similarly, the true $p_{3->4}$ would be smaller than the measured value. However, three observations suggest that it is unlikely for the two isoforms to have different fibril core sizes. First, tau filaments obtained from multiple AD brains have been shown by cryoEM

to have the same β-sheet core, without signs of structural heterogeneity[4,5]. Second, the 2D hNH dipolar and INEPT spectra measured here (Fig. 4) show similar spectral peak patterns for the mixed isoform samples and the single isoform samples. Thus, the AD tau seeds are able to propagate by either isoform to develop the same structure. Third, the AD tau seeds are well amplified in our current study, which would be difficult to achieve if a monomer of the opposite isoform is added to the growing fibril with a different core size than the current isoform.

The moderate over-incorporation of 4R over 3R tau into the AD tau filament is consistent with fluorescence data of mammalian cells expressing 4R and 3R tau repeat domains. These data showed that both 4R and 3R tau are aggregated by AD brain extracts, but overexpression of 4R tau increased the total amount of the aggregates more than 3D tau did[16]. It is also consistent with the observation that 4R tau isoforms have faster fibril nucleation rates and elongation rates than 3R tau isoforms in vitro[35,36].

The isoform-independent propagation of the AD tau structure and pathological properties is noteworthy. The R2 domain contains a fibrillary hexapeptide motif like the R3 domain[37–39]. Thus, if a series of 4R tau monomers is added to the elongating filament, we would expect their R2 domains to begin to form cross-β hydrogen bonds and join the rigid core. This should gradually lead to a new fibril structure that prefers to incorporate 4R tau instead of 3R tau. The 2D NMR spectra measured here indicate that the AD tau is able to avoid this fate, even when only 4R tau monomers are added to the seed. The molecular mechanism for this avoidance is still unclear. It is possible that the posttranslational modification pattern of AD tau favors 4R tau incorporation[36] as well as predisposing the seeded structure to prevent the incorporation of the R2 domain into the β-sheet core. The AD tau fold may have evolved over its slow replication phase[40] to adapt to the environment of the AD brain[41]. The fluent molecular mixing of both 4R and 3R tau isoforms and the preservation of the β-sheet conformation regardless of the isoform could potentially be an adaptive trait to optimize prion growth and propagation. The significant amplification of the morphology and toxicity of AD tau filaments shown here opens new avenues for future characterization of the molecular structure of AD tau using a wide range of biophysical methods.

## Methods

**Enrichment of AD tau**. AD tau was sequentially extracted from different regions of patient brains as previously described[18]. Patients were autopsied at the Center for Neurodegenerative Disease Research (CNDR) in accordance with state laws and protocols approved by the University of Pennsylvania. Informed consent was obtained from the patients' next of kin. All patients had a neuropathological diagnosis of AD at Braak stage VI. Brain tissues were obtained from the Integrated Neurodegenerative Disease Database (INDD), including six females and four males (Supplementary Table 1) whose age ranges from 55 to 92 years. Postmortem interval ranges from 4 to 23 h. As this study only uses proteins from postmortem tissues and does not involve living human subjects, it is not subject to IRB approval (45 CFR 46). All cases were screened to exclude the comorbidities of Lewy body, TDP-43, and glial tau pathologies based on immunohistochemical staining. Enzyme-linked immunosorbent assay and immunoblot measurements of tau, α-synuclein, amyloid-beta, and total protein concentrations are shown in Supplementary Table 2.

**Expression and purification of perdeuterated 0N4R and 0N3R Tau**. Genes for 0N4R and 0N3R tau were cloned into a pET-28a vector and transfected into *E. coli*. BL21 (DE3) competent cells (Novagen) as described previously[22,23]. Colonies from freshly transformed LB Agar plates were used to inoculate 50 ml starter cultures of H₂O LB media. All cultures were grown at 37 °C under 250 rpm shaking and contained 50 μg/ml kanamycin. After overnight growth, 4 ml of the H₂O LB starter culture was added to 25 ml of fresh D₂O LB media. This culture was grown to an $OD_{600}$ of ~1.0 before it was added to 100 ml of fresh D₂O M9 media containing suitably labeled glucose at 2 g/l and ammonium chloride at 0.5 g/l. The M9 culture was grown to an $OD_{600}$ of ~1.0 before it was added to 400 ml of fresh D₂O M9 media. When the 500 ml of D₂O M9 reached $OD_{600}$ ~1.0, protein expression was induced with 0.5 mM IPTG. At this time an additional 1 g/l glucose was added. After 3–5 h of induction, cells were harvested by centrifugation and cell pellets were

frozen. The D₂O M9 media contained 0.5 g/l of NH₄Cl or ¹⁵NH₄Cl as required, and a final concentration of 3 g/l ²H-labeled glucose or ²H, ¹³C-labeled glucose (Cambridge Isotope Laboratories) as required. The media contained standard M9 salts, 0.1 mM CaCl₂, 2 mM MgSO₄, 50 mg/l kanamycin, 1× lyophilized MEM vitamins, and 0.2× Studier Metals.

0N4R and 0N3R tau were purified as described before[22,23]. Briefly, frozen cell pellets were thawed and homogenized by vortexing on ice in ice-cold lysis buffer. The buffer contained 20 mM Na₂HPO₄ (pH 6.8), 50 mM NaCl, 1 mM EGTA, 0.2 mM MgCl₂, 5 mM DTT, 0.1 mg/ml lysozyme, and 1× cOmplete™ protease inhibitor cocktail (Roche) per 50 ml lysis buffer. Lysis was performed on ice with a probe sonicator, after which cell lysate was boiled for 20 min in a water bath, then centrifuged at 15,000g for 40 min to remove cell debris and aggregated protein. The supernatant was purified with a cation-exchange column (self-packed with SP Sepharose Fast Flow resin; GE healthcare), then further purified using reverse phase HPLC (Agilent Zorbax 300SB-C3 column, 21.2 × 250 mm, 7 μm particle size) using an acetonitrile gradient of 5–50% in 50 min. HPLC fractions were pooled and lyophilized to yield ~10 mg/l of perdeuterated tau. The yields were similar between 0N4R and 0N3R tau.

**Quantification of Tau monomers for fibrillization and seeding reactions**. To precisely control the amount of tau monomers added to the fibrillization mixtures, we dissolved purified and lyophilized ²H, ¹⁵N (DN) or ²H, ¹³C (DC)-labeled tau in ice-cold phosphate buffer saline (PBS) buffer at 10–15 mg/ml in the absence of DTT. Monomer concentration was measured with a NanoDrop Lite (Thermo Fisher). This high concentration (10–15 mg/ml) of tau monomers was used for quantification because the tau extinction coefficients are predicted to be low: $A_{280}$ 0.186 for 1 mg/ml 0N4R tau and 0.203 for 1 mg/ml 0N3R tau. These values do not account for increased isotopic mass. Absorbances were measured in triplicate and averaged. These gave an estimated uncertainty of ~1% in the reported ¹³C:¹⁵N monomer ratios in the reaction mixture.

**Heparin-induced fibrillization of Tau**. Heparin-induced tau fibrils were prepared as described before[22]. Ten micromolar DN- or DC-labeled 0N4R tau monomers (e.g., 0.4 mg/ml total 0N4R tau) and 1 mM DTT were added to pH 7.4 PBS buffer and bath sonicated for 1 min. To this solution we added 0.125 mg/ml heparin (Santa Cruz Biotech, sc203075, 8000–25,000 Da), briefly vortexed to mix, then split the solution into 1.5 ml Eppendorf tubes. The tubes were shaken at 37 °C and 1220 rpm with 2 mm orbit for 3 days. After this, the solutions were ultracentrifuged in a TLA-55 rotor at 100,000g for 1 h, and the fibrils were collected by discarding most of the supernatant. The fibrils were resuspended, combined, and then subjected to a second ultracentrifugation step to obtain the total pellet.

**Seeded amplification of AD Tau fibrils**. AD tau seeds were amplified in solutions containing final concentrations of 4 μM seeds and 36 μM total recombinant tau, along with 2 mM DTT, 5 mM PMSF, and 1% v/v of a protease inhibitor cocktail (1% w/v each of pepstatin, leupeptin, TPCK, TLCK, and trypsin inhibitor, 0.1 M EDTA), in PBS buffer. Initially, AD seeds, DTT, and protease inhibitors for 4 × 100 μl reactions were combined into a single 0.2 ml flat capped PCR tube (Fisher Scientific 14230225), with a volume of ~120 μl. The seed mixture was heat treated at 56 °C in a thermocycler for 30 min, and then sonicated by securing tubes at the antinodes of a Branson 2200 bath sonicator for 30 min. When sonicating, the tubes were secured in place within individual flotation devices which were pinned to rods taped to the sonicator. The exact positions of the tubes were adjusted for each use to maximize sonication force. The seed mixture was then split evenly into four PCR tubes, into which the appropriate amount of recombinant tau solution and PBS buffer were added to bring the final volume to 100 μl. When 4R and 3R tau monomers were mixed, each monomer was added to 18 μM final concentration; otherwise 36 μM of a single monomer was used. Reactions were wrapped in parafilm and shaken at 37 °C and 1220 rpm with 2 mm orbit for 7 days. Aliquots were taken immediately before and after 7 days of shaking as day 0 and day 7 samples, respectively. Monomer-only controls used PBS buffer instead of AD tau seeds. SDS-PAGE and neurotoxicity assays were conducted in duplicate for each AD-tau-seeded sample at day 0 and day 7. These duplicates were taken from reactions that used separately heat-treated and sonicated seed and protease inhibitor mixtures, although all protease inhibitors originally came from one stock, and all seeds came from one pool of AD brain material. After taking aliquots for biochemical assays, seeding reactions were pooled and fibrils were collected by ultracentrifugation in a TLA-55 rotor at 100,000g for 1 h.

**Sedimentation assay of fibril samples**. Sedimentation assays were conducted by mixing a 1 μl aliquot of fibrillization reaction mixture with 19 μl of PBS (0.1% sarkosyl, w/v) and ultracentrifuging for 30 min at 100,000g (Optima, Beckman Coulter) as previously described[19]. The supernatant was carefully removed, and the pellet was resuspended with 20 μl PBS. Lysates were mixed with loading buffer for immunoblots. Samples were separated using a 10% SDS-PAGE gel.

Tau species were revealed using antibodies 17025 (made in-house) and PHF1 (a gift from Dr. Peter Davis). The 17025 antibody was raised with recombinant human tau; thus, it has a higher affinity for recombinant tau than AD brain tau. PHF1 is more specific to the S396/S404 phosphorylated tau; thus, the bands in

PHF1 blots (Fig. 1b) mostly represent hyperphosphorylated AD tau seeds. As there are six isoforms of human tau and high molecular-weight tau species are present in the AD seeds, the banding pattern of PHF1-tau is more complex than the pattern of recombinant tau (17025-tau).

Because the AD tau seeds (PHF-1 positive bands) were sonicated before being added to the seeding reactions, ultracentrifugation on day 0 at 100,000g for 30 min in the sedimentation assay could not completely sediment the insoluble AD tau seeds, and a significant amount of tau seeds stay in the supernatant. After 7 days of seeding reactions, tau filaments were elongated and became easier to be pelleted, resulting in a shift in the amount of AD tau to the pellet fraction. But the total amount of AD tau in the supernatant and pellet was similar before and after the reaction. Thus, there is no evidence of kinases in the insoluble tau seeds.

**Transmission electron microscopy (TEM) of Tau fibrils**. After fibril growth, 1 µl aliquots of fibrillization reactions were diluted fivefold using deionized water and adsorbed onto freshly glow-discharged, 200 mesh Formvar/carbon-coated copper grids (Ted Pella) for 60 s, blotted and washed with deionized water twice, and stained with 0.5% (w/v) uranyl formate for 20 s. TEM images were measured on an FEI Tecnai G2 Spirit TWIN electron microscope. Raw image files were processed and exported using ImageJ 1.53a. The dimensions of the fibrils were sampled manually using the built-in measurement tool of ImageJ. At least five different particles from two or more different TEM images were sampled and a total of at least 10 readings of width and crossover length were averaged to give the statistics shown in Supplementary Table 3.

**Pathogenicity test in primary neurons**. CD1 wild-type mice were used to generate primary hippocampal neurons. Female pregnant mice were purchased from Charles River. Animals were kept in microisolator cages at a temperature between 18 and 23 °C with 40–60% humidity, with 12 h light on and 12 h light off. All animal care and experimental protocols were approved by the University of Pennsylvania's Institutional Animal Care and Use Committee. CD1 mouse cortices and hippocampi were dissected at embryo days 16–18 and dissociated with papain (Worthington Biochemical Corporation). Neurons were resuspended in neural basal medium (Gibco, 21103) with 2% B27 (Gibco), 1× Glutamax (Gibco), and 1× penicillin/streptomycin (Gibco). Plates or coverslips were coated with poly-D-lysine (0.1 mg/ml, Sigma-Aldrich) in borate buffer (0.05 M boric acid, pH 8.5) overnight at room temperature. Cells were plated at a density of 50,000 cells/cm$^2$ for 96- or 384-well plates. When plating, 5% fetal bovine serum (FBS) was added to the cell suspension. The plating medium was replaced with neural basal medium without FBS at day 1 in vitro (DIV1). At DIV7, the plating medium was replaced with condition medium containing 1:1 ratio of old and fresh medium. AD tau seeds or seeded reactions were first diluted in PBS to the desired tau concentrations, and then added to the cells and incubated for 14 days until DIV21. The dose of tau fibrils was converted based on cell densities (µg tau/10$^6$ cells). For activity measurements, if not otherwise specified, 1 µg tau/10$^6$ cells were used for 100% seeds (100% pathogenic tau), 0.1 µg tau/10$^6$ cells for 10% seeds control, and 0.01 µg tau/10$^6$ cells for 1% seeds control. All amplified tau variant doses were based on the total tau concentration at the start of the in vitro reaction. One-way ANOVA followed by Tukey post hoc test was conducted using 3–4 biological repeats. This number of repeats was utilized based on previous experience in similar studies obtaining statistically significant results.

**Immunocytochemistry**. Cells were washed four times with PBS and extracted with 1% hexadecyltrimethyl–ammonium bromide (HDTA; Sigma-Aldrich) for 10 min following 10 min of fixation with 4% paraformaldehyde and sucrose in PBS. Fixed cells were stained with primary antibodies overnight at 4 °C, and then with Alexa Fluor-conjugated secondary antibodies (Thermo Fisher Scientific) for 2 h at room temperature. Images were randomly taken using an InCell Analyzer 2200 microscope and quantified for the antibody immunoreactivity. Tau pathology was quantified as the area of occupancy × intensity/DAPI count.

**Solid-state NMR experiments**. All AD-tau seeded tau fibrils and heparin-fibrilized tau samples were packed into Bruker 1.3 mm magic-angle-spinning (MAS) rotors using a custom-designed ultracentrifuge packing tool. Fibril pellets were resuspended in ~120 µl of their own supernatant, loaded into the packing tool, and then spun on a Beckman Optima XL-80 with a SW60 Ti rotor at 311,000g for 16 h at 10 °C. For AD-tau-seeded samples, each MAS rotor contained ~4 mg hydrated material, a quarter of which was the dry mass of isotopically labeled recombinant tau. For heparin-fibrilized tau samples, ~2 mg hydrated material was packed into the rotor, in which ~1 mg was the dry mass of the protein.

Solid-state NMR experiments were conducted on a Bruker Avance III HD 600 MHz (14.1 T) spectrometer and an Avance II 800 MHz (18.8 T) spectrometers using 1.3 mm HCN MAS probes. $^1$H chemical shifts were referenced externally to sodium trimethylsilylpropanesulfonate (DSS) at 0 ppm. $^{13}$C and $^{15}$N chemical shifts were referenced externally using the tripeptide N-formyl-Met-Leu-Phe-OMe (f-MLF). The Met Cε peak was calibrated to 14.0 ppm on the tetramethylsilane (TMS) scale while the Phe $^{15}$N peak was calibrated to 110.09 ppm on the liquid ammonia scale. Most experiments were conducted under 20 kHz MAS, except for the 2D dipolar hNH and 3D hCANH and hCA(co)NH experiments, which were

conducted under 55 kHz MAS. Sample temperatures were kept at 285 or 297 K, which was estimated based on the water $^1$H chemical shift ($\delta_{H2O}$) of 4.89 or 4.77 ppm, respectively using the equation $T_{eff}$ (K) = 96.9 × (7.83 − $\delta_{H2O}$)[42]. Typical radiofrequency (rf) field strengths were 65–90 kHz for $^1$H, 62.5 kHz for $^{13}$C, and 45–50 kHz for $^{15}$N. In all $^1$H-detected experiments, water suppression was carried out using the MISSISPPI sequence with a $^1$H rf field strength of 15 kHz[43]. For REDOR experiments under 20 kHz MAS, 130 kHz $^1$H SPINAL-64 decoupling was applied during the $^{15}$N–$^{13}$C REDOR period, while 3.6 kHz $^{15}$N and $^{13}$C WALTZ-16 decoupling was applied during $^1$H detection. For 2D hNH experiments under 55 kHz MAS, 10 kHz $^1$H, $^{15}$N, and $^{13}$C WALTZ-16 decoupling were applied.

One-dimensional (1D) NMR spectra for quantifying the amounts of $^{15}$N-labeled and $^{13}$C-labeled tau in the samples were conducted under 20 kHz MAS. Quantitative $^{13}$C direct polarization (DP) spectra were measured at 285 K using a recycle delay of 5 s and 2048 scans. 1D J-hnH spectra were measured to quantify the amount of $^{15}$N spins in the dynamic portion of the protein. These spectra were measured at 297 K. 1D dipolar hnH spectra were measured at 285 K to quantify the amount of $^{15}$N spins in the rigid core. The experiment used $^1$H–$^{15}$N and $^{15}$N–$^1$H cross-polarization (CP) contact times of 1.25 and 0.4 ms, respectively. The spectra were measured with 256 scans and a recycle delay of 1.5 s.

Molecular mixing between $^{15}$N-labeled and $^{13}$C-labeled tau was detected using the $^1$H-detected $^{15}$N-$^{13}$C REDOR experiment at 285 K. A $^{15}$N–$^{13}$C REDOR period with a variable dipolar recoupling time of 0.1 to 25 ms was inserted before the MISSISPPI period in a dipolar hnH pulse sequence (Fig. 3c). A total of 28 mixing times were measured per sample. The control spectrum ($S_0$) without $^{13}$C pulses and the dephased spectrum (S) with $^{13}$C pulses were measured back-to-back for each mixing time to avoid spectrometer drift. In total, 1536–2048 repeated scans were coadded for each mixing time. This signal averaging was separated into 6–8 blocks of the complete set of mixing times to allow re-optimization of experimental conditions between blocks. A full REDOR curve for each sample was measured in 3 days.

2D $^{15}$N–$^1$H correlation spectra were measured to compare the AD-tau seeded samples and heparin-fibrilized 0N4R tau sample. The 2D dipolar hNH spectra selectively detect the rigid core. The spectra of heparin-fibrilized 0N4R tau and [AD]4R$^{NC}$ were measured on the 800 MHz NMR, while the other samples were measured on the 600 MHz NMR. Except for the magnetic field difference, all experiments were conducted under the same conditions, including 55 kHz MAS, a sample temperature of 285 K, and 1.25 ms $^1$H–$^{15}$N CP and 0.4 ms $^{15}$N–$^1$H CP contact times. The 2D spectra were measured with 38 ms $^1$H acquisition time and up to 30 ms $^{15}$N evolution. Each 2D spectrum took 2–3 days of signal averaging. 2D INEPT (J-hNH) spectra were measured to observe the signals of isotropically mobile residues. The spectra were measured on the 600 MHz NMR at 297 K, with 92 ms $^1$H acquisition and up to 70 ms $^{15}$N evolution time. Each spectrum took ~2 days of signal averaging. All 2D spectra were processed using QSINE = 3 apodization.

3D hCANH and hCA(co)NH correlation spectra were used to assign part of the backbone of the rigid core of heparin-fibrilized 0N4R tau. Both experiments started from 2 ms $^1$H–$^{13}$C CP and underwent up to 4.8 ms $^{13}$C evolution. Intra-residue Cα–N polarization transfer was accomplished by 8 ms SPECIFIC CP in hCANH, while inter-residue CO-N transfer was carried out using consecutive 8 ms Cα-CO DREAM and 8 ms CO-N SPECIFIC CP. The $^{15}$N evolution time was up to 10.5 ms, and 0.4 ms $^{15}$N-$^1$H CP transferred the polarization to $^1$H for detection. Both 3D spectra were processed using GM apodization with LB = 15 Hz and GB = 0.07 and forward linear prediction. Detailed experimental parameters can be found in Supplementary Table 5.

**Two-state Markov model of 4R and 3R Tau mixing with probabilities $p_{3\to4}$ and $p_{4\to3}$**. We consider a parallel-in-register cross-β fibril composed of a mixture of 4R and 3R tau. We assume that the fibril has a constant 4R:3R tau molar ratio over its entire length, and the probability $p_{4\to3}$ that a 4R tau is followed by a 3R tau is the same as the probability that a 4R tau is preceded by a 3R tau (Fig. 3a). Then the probability that the 4R tau is followed or preceded by another 4R tau is $p_{4\to4} = 1-p_{4\to3}$. The probabilities $p_{3\to3}$ and $p_{3\to4}$ are defined analogously.

We next define the mole fraction of 4R tau in one part of the fibril as $\chi_4$. The mole fraction of 3R tau in the same part of the fibril is then $\chi_3 = 1 - \chi_4$. If we shift the portion of the fibril under consideration by one monomer, then the new mole fraction $\chi_4'$ of 4R tau is

$$\chi_4' = \chi_4 p_{4\to4} + \chi_3 p_{3\to3} \tag{1}$$

Because the 4R:3R tau concentration ratio is assumed to be constant throughout the fibril, this new mole fraction must be equal to the original mole fraction, $\chi_4' = \chi_4$. From this we obtain

$$\chi_4 = \frac{p_{3\to4}}{p_{3\to4} + p_{4\to3}} \tag{2}$$

Therefore, the measured mixing probabilities encode the mole fractions of 4R and 3R tau. These mole fractions are equivalent to the global incorporation levels of 4R and 3R tau, which reflect the preference of AD PHF tau for the two isoforms.

The preference of AD-tau seeded 4R and 3R tau to separate like block copolymers or alternate along the fibril axis can be described independently of the isoform incorporation level. To quantify this property, we define a mixing quotient,

which is analogous to the equilibrium constant of mixing:

$$Q = \frac{p_{3\rightarrow4}p_{4\rightarrow3}}{p_{4\rightarrow4}p_{3\rightarrow3}} \qquad (3)$$

This mixing quotient reflects the tendency of a like-interface (4–4 or 3–3) to dissociate to form an unlike interface (4–3 or 3–4). If $Q$ is much larger than 1, then there is a strong preference for 4R and 3R to alternate. A perfectly alternating 4R–3R–4R–3R fibril has an infinite $Q$. Conversely, if $Q$ is much less than 1, then there is a strong preference for 4R–4R and 3R–3R contacts, i.e., a homotypic preference. In this block copolymerization case, $p_{4\rightarrow4}$ and $p_{3\rightarrow3}$ are much larger than $p_{3\rightarrow4}$ and $p_{4\rightarrow3}$. If there is neither preference for alternation nor neither for block copolymerization, then $p_{3\rightarrow4} = p_{4\rightarrow4} = \chi_4$, and $Q = 1$.

Therefore, assuming there are no additional correlations between non-neighboring chains, $p_{3\rightarrow4}$ and $p_{4\rightarrow3}$ completely characterize the mixing properties of a uniform fibril. The parameters $Q$ and $\chi_4$ are alternative and equivalent expressions of these two probabilities. $p_{3\rightarrow4}$ and $p_{4\rightarrow3}$ can be measured by $^{15}N$–$^{13}C$ REDOR, giving experimental access to the mixing quotient and the global incorporation level of 4R tau.

**Extraction of $p_{4\rightarrow3}$ and $p_{3\rightarrow4}$ from REDOR curves.** The probability $p_{4\rightarrow3}$ and $p_{3\rightarrow4}$ in AD seeded samples were extracted by comparing the $^{15}N$-$^{13}C$ REDOR curves of [AD]4R$^N$3R$^C$ and [AD]3R$^N$4R$^C$ with the REDOR curves of heparin-fibrillized calibration samples with known ratios of $^{15}N$-labeled and $^{13}C$-labeled 0N4R tau (Fig. 3e). The REDOR data of the AD-seeded mixed 4R/3R tau were measured in duplicate using separately seeded samples (#1 and #2 for each labeling scheme). The REDOR curves of the two replicates of [AD]4R$^N$3R$^C$ were averaged, then the $p_{4\rightarrow3,\,i}$ value at each mixing time $i$ between 8 and 25 ms was linearly interpolated from the REDOR calibration curves for $p_{N\rightarrow C} = 0.3$ and $p_{N\rightarrow C} = 0.4$ according to

$$p_{4\rightarrow3,i} = \frac{\left(\frac{S}{S_0}_{N4RC3R,i} - \frac{S}{S_0}_{40/60,i}\right)}{\left(\frac{S}{S_0}_{30/70,i} - \frac{S}{S_0}_{40/60,i}\right)} * (30-40) + 40 \qquad (4)$$

These individual $p_{4\rightarrow3,i}$ values were averaged to give the final $p_{4\rightarrow3}$ value. The error bar in the $p_{4\rightarrow3}$ value is propagated from the spectral signal-to-noise ratios of the calibration spectra and the AD-tau seeded spectra using the following equation :

$$\varepsilon(p_{4\rightarrow3}) = 10 * \left[ \frac{1}{n} \sum_i^n \left( \left( \frac{\varepsilon_{N4RC3R,i}}{\frac{S}{S_0}_{30,i} - \frac{S}{S_0}_{40,i}} \right)^2 + \left( \frac{\left( \frac{S}{S_0}_{N4RC3R,i} - \frac{S}{S_0}_{40,i} \right)* \varepsilon_{40,i}}{\left( \frac{S}{S_0}_{30,i} - \frac{S}{S_0}_{40,i} \right)^2} \right)^2 + \left( \frac{\left( \frac{S}{S_0}_{N4RC3R,i} - \frac{S}{S_0}_{40/60,i} \right)* \varepsilon_{70,i}}{\left( \frac{S}{S_0}_{30/70,i} - \frac{S}{S_0}_{40/60,i} \right)^2} \right)^2 \right) \right]^{1/2} \qquad (5)$$

Here $\varepsilon$ is the spectral noise obtained from the signal-to-noise ratio of the REDOR spectra and $n = 18$ for the number of REDOR mixing times from 8 ms to 25 ms.

The $p_{3\rightarrow4}$ value was similarly extracted by comparing the REDOR curves of [AD]3R$^N$4R$^C$ samples with the REDOR data of the calibration samples for $p_{N\rightarrow C} = 0.5$ and $p_{N\rightarrow C} = 0.6$. Errors in $p_{3\rightarrow4}$ and $p_{4\rightarrow3}$ were propagated to errors in the calculated $\chi_4$ and $Q$ following Gauss' law of error propagation.

Among the measured REDOR calibration curves from the heparin-fibrillized 0N4R tau samples, the curve of the 100% $^{15}N$-labeled 0N4R tau without $^{13}C$-labeled protein already exhibits dipolar dephasing to 0.84 at 25 ms (Fig. 3e and Supplementary Fig. 3a). This indicates that natural abundance $^{13}C$ spins cause non-negligible dephasing to $^{15}N$. This natural abundance effect is approximately corrected by dividing all measured REDOR intensities by the intensities of the 100% $^{15}N$-labeled sample (Supplementary Fig. 3b).

**Estimation of Tau monomer fractions in the seeded samples by 1D NMR spectra.** To provide additional verification of the overall incorporation level of 4R and 3R tau in the fibril, we quantified the intensities of 1D $^1H$-detected $^{15}N$ NMR spectra as well as $^{13}C$ spectra. Quantitative $^{13}C$ DP NMR spectra were measured using a 5 s recycle delay to quantify the amount of total $^{13}C$-enriched tau monomers in the fibrils. 1D CP-hnH spectra were used to quantify the amount of the rigid portions of $^{15}N$-enriched tau, while J-hnH spectra were used to quantify the amount of the mobile portions of $^{15}N$-enriched tau (Supplementary Fig. 5). The 5 s DP $^{13}C$ spectra were integrated between 200 and 0 ppm, while the hnH spectra were integrated between 12 and 6 ppm. The intensity ratios of the J-hnH spectra and the ratios of the CP-hnH spectra were averaged. As the double-labeled [AD]4R$^{NC}$ contains an equal amount of $^{13}C$ and $^{15}N$ tau, normalizing the $^{13}C$ and $^{15}N$ intensities from the mixed-labeled samples to the [AD]4R$^{NC}$ intensities allows quantification of the $^{13}C$: $^{15}N$ molar ratios, and hence the 4R:3R tau molar ratios. [AD]4R$^N$3R$^C$ sample #1 was found to contain 60.0 ± 3.0% 4R tau, whereas sample #2 contained 62.6 ± 3.0% 4R tau (Supplementary Table 4). Following a similar procedure, we found that the two duplicates of [AD]3R$^N$4R$^C$ contained 60.8% ± 3.0% and 62.1% ± 3.0% of 4R tau, respectively. The 3.0% uncertainty was estimated based on heparin-fibrillized 0N4R tau calibration samples, where the $^{15}N$-labeled 4R monomer fraction derived from the 1D spectra deviated from the added fraction by 3.0%. We attribute this uncertainty to multiple polarization transfer steps in the J-hnH and

CP-hnH experiments. The actual uncertainty in the monomer concentration is 1.0% based on the nanodrop reading.

**REDOR simulations using SpinEvoluton and second moment analysis.** To assess whether the experimentally measured $^{15}N$–$^{13}C$ REDOR curves of in vitro calibration samples are in good agreement with the known cross-β fibril geometry, and to evaluate the accuracy of the mixing probability measurement using this approach, we simulated the REDOR curves for varying ratios of $^{13}C$-labeled protein and $^{15}N$-labeled protein in an idealized cross-β fibril. The simulations were carried out in three ways, using (1) the SpinEvolution software, (2) the second-moment analysis for up to 9 $^{13}C$ spins, and (3) the second-moment analysis for up to 48 $^{13}C$ spins. These different approaches are compared (Supplementary Fig. 3c–e, f–h) to account for the large number of $^{13}C$ spins, both labeled and natural abundance ones, near each observed $^{15}N$ spin.

All three simulations consist of superpositions of multiple REDOR curves weighed by the probabilities of $^{13}C$ and $^{15}N$ mixing in a cross-β hydrogen-bonded fibril. Denoting the $^{13}C$:$^{15}N$ mixing ratio in the 0N4R tau calibration samples as $p_{N\rightarrow C}$, each of the four packing modes in Fig. 3b occurs with a well-defined probability: $(1-p_{N\rightarrow C})^2$ for the N–N–N packing mode, $p_{N\rightarrow C}(1-p_{N\rightarrow C})$ for the N–N–C packing mode, $(1-p_{N\rightarrow C})p_{N\rightarrow C}$ for the C–N–N packing mode, and $(p_{N\rightarrow C})^2$ for the C–N–C packing mode. The REDOR dephasing of the individual packing modes was weighted by these for $p_{N\rightarrow C}$ values to give the total REDOR dephasing curves for each $^{13}C$:$^{15}N$ mixing ratio. To obtain the closest $^{15}N$–$^{13}C$ distances in a standard cross-β hydrogen-bonded fibril, we used I308 in the AD PHF tau structure (PDB: 5O3L) as the reference $^{15}N$ spin and extracted its distances to the nearest $^{13}C$ spins in the two neighboring chains in PyMOL.

Simulations using SpinEvolution[27] explicitly account for all nuclear spin interactions but are restricted to a maximum of 9 $^{13}C$ spins for each $^{15}N$ spin. In these SpinEvolution simulations, we neglect the natural abundance (1.1%) $^{13}C$ nuclei in the central $^{15}N$-labeled monomer. The coordinates of $^{13}C$ nuclei relative to the I308 $^{15}N$ were extracted in PyMOL. The internuclear distance $r_{NC}$ ranges from 4.1 to 5.8 Å. Specifically, in the N–N–N packing mode, no $^{13}C$ spin was considered and hence there is no REDOR dephasing. In the N–N–C packing mode, all five $^{13}C$ spins within 5.8 Å were simulated (Fig. 3b). In the C–N–N packing mode, all eight $^{13}C$ spins within 5.8 Å were simulated. In the C–N–C packing mode, all nine $^{13}C$ spins within 5.5 Å were simulated (Supplementary Fig. 3i). $^{13}C$–$^{15}N$ and $^{13}C$–$^{13}C$ dipolar couplings were calculated internally by the software. The chemical shift anisotropy (CSA) of $^{15}N$ was neglected. All aliphatic $^{13}C$ nuclei were given a CSA anisotropy $\delta = \delta_{zz} - \delta_{iso}$ of 30 ppm, an asymmetry parameter $\eta = (\delta_{yy} - \delta_{xx})/\delta$ of 0, where $\delta_{xx}$, $\delta_{yy}$, and $\delta_{zz}$ are the principal values of the CSA and $\delta_{iso}$ is the average of the three principal values. All carbonyl $^{13}C$ nuclei were given a $\delta$ of 170 ppm and $\eta$ of 0. The orientation of the CSA tensors was set to be random relative to the dipolar vector. Random coil chemical shifts were used as the isotropic chemical shift of each atom[44]. The $^1H$ Larmor frequency was set to 600 MHz, and the MAS frequency was set to 20 kHz. The rf carrier frequency for $^{13}C$ and $^{15}N$ channel were set to 100 and 120 ppm, respectively, and 8 μs rectangular 180° pulses were used for both channels. Powder averaging was carried out using the rep30 set of Euler angles based on the REPULSION power averaging scheme[45]. Options "-dw123 -re" were applied. The $^{15}N$ signal was simulated for REDOR mixing times of 0–50 ms in 0.1 ms steps. The SpinEvolution code used in the simulation can be found in the Source Code.

To evaluate the effects of more than nine $^{13}C$ spins to REDOR dephasing, we conducted a second-moment analysis for the same set of $^{13}C$ spins used for SpinEvolution and for up to 48 $^{13}C$ spins within a radius of 8.8 Å to the I308 $^{15}N$ in neighboring monomers (Supplementary Fig. 3i). Following Hirschinger[28], we calculated the second moment of $n$ $^{13}C$–$^{15}N$ dipolar couplings by summing the squares of the individual couplings and dividing by 5,

$$M_2 = \frac{1}{5} \sum_i^n \left( \frac{3000\text{Hz}}{r_i^3} \right)^2. \qquad (6)$$

Here 3000 Hz is the $^{13}C$–$^{15}N$ dipolar coupling constant for a 1 Å distance and the internuclear distance $r$ is expressed in the unit of Å. The REDOR dephasing curve for these $n$ $^{13}C$–$^{15}N$ dipolar couplings has a Gaussian shape:

$$\frac{S}{S_0} = \exp\left(-4M_2 t^2/3\pi^2\right) \qquad (7)$$

This second moment analysis ignores $^{13}C$ finite pulse effects, $^{13}C$ chemical shifts, and $^{13}C$–$^{13}C$ dipolar couplings. The second moment analysis was conducted using Microsoft Excel 2021, with equation embedded in the Source Data.

Supplementary Figure 3 shows that the SpinEvolution simulations and second-moment analysis produce REDOR dephasing curves that deviate in opposite ways from the measured curves after $^{13}C$ natural abundance correction. The 9-spin SpinEvolution simulations result in slightly higher $S/S_0$ values from the experimental values, whereas the 48-spin secondary moment analysis result in lower $S/S_0$ values than the experimental values (Supplementary Fig. 3f, h). The 9-spin second-moment analysis approaches the experimental data the closest (Supplementary Fig. 3g). The faster dephasing of the simulated REDOR curves with 48 nearest $^{13}C$ spins compared to dephasing by 9 $^{13}C$ spins indicate that

$^{15}$N–$^{13}$C distances between 5.8 and 8.8 Å cannot be neglected. Thus, SpinEvolution simulation for up to nine $^{13}$C spins is an underestimate. On the other hand, the explicit SpinEvolution simulation with nine $^{13}$C spins decays more slowly than the second moment analysis for nine spins, indicating that other spin interactions, including $^{13}$C–$^{13}$C dipolar couplings and chemical shifts, slow down the $^{15}$N–$^{13}$C dipolar dephasing. This means the second-moment analysis of 48 $^{13}$C spins overestimates the true decay rate, by removing, for example, the effect of $^{13}$C–$^{13}$C dipolar coupling truncating the $^{13}$C–$^{15}$N difference tensors during the REDOR evolution. Therefore, neither quantum-mechanical simulations of a limited number of spins and second-moment analysis of a large number of spins without all spin interactions fully account for both long-range $^{13}$C–$^{15}$N distances and all nuclear spin interactions. Due to these limitations, we rely on the experimentally measured calibration REDOR curves to extract the mixing probabilities of AD-tau seeded filaments accurately.

**Mixed fibril construction and modeling of 4R/3R tau mixing**. We constructed a cross-β filament containing $360 \times 2$ monomers of tau using the AD PHF tau cryo-EM structure (PDB: 5O3L) (Fig. 3f, g). The $5 \times 2$ monomers in the original PDB structure were duplicated, then the duplicate was translated by 4.7 Å per monomer along the fibril axis, and rotated around the fibril axis by 1° per monomer. The duplicate was then merged with the original PyMOL object. This process was repeated until the filament contained $360 \times 2$ monomers.

A python script was written to simulate a filament of 360 tau monomers with 4R and 3R mixed according to the measured $p_{3\to4}$ and $p_{4\to3}$ values. The simulation was repeated until the fraction of 4R tau followed by a 3R tau is $37 \pm 1\%$, and the fraction of 3R tau followed by a 4R tau is $56 \pm 1\%$. This was done independently for both protofilaments. This script can be found in Source Code.

To produce statistical models of 100-monomer filaments with varying $Q$ and $\chi_4$ values (Supplementary Fig. 6), we first converted $Q$ and $\chi_4$ to $p_{3\to4}$ and $p_{4\to3}$ values, and then randomly simulated the fibrils using these probabilities. These short simulations have high variance of observed $Q$ and $\chi_4$; thus, the 100-monomer fibrils were repeatedly simulated until both $Q$ and $\chi_4$ were within 10% and 5% of their desired values, respectively. This script can be found in the Source Code.

**Reporting summary**. Further information on research design is available in the Nature Research Reporting Summary linked to this article.

## Data availability
The REDOR NMR intensities and pathological activities data generated in this study are provided as a Source Data file. NMR spectra are available from M.H. upon request. The AD PHF tau cryo-EM structure used for the construction and modeling of 4R/3R tau mixing is publicly available following the link [https://doi.org/10.2210/pdb5O3L/pdb].

## Code availability
The SPINEVOLUTION REDOR simulation codes, the python code for building a fibril model with desired mixing probabilities, and the python code for simulating tau fibrils with different $Q$ and $\chi_4$ values, are included in the Source Code.

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

## Acknowledgements

This work was supported by NIH grants AG059661 to M.H. and AG17586 to V.M.-Y.L. A.J.D. is supported by an NIH Ruth L. Kirschstein Individual National Research Service Award (F31AG069418) and H.X. is supported by a grant from the BrightFocus Foundation. This study made use of NMR spectrometers at the MIT-Harvard Center for Magnetic Resonance, which is supported by NIH grant P41 GM132079. We thank Prof. Haifan Wu for fibrilling the heparin CDN-labeled tau sample. We thank the patients and their families for donating brain tissues.

## Author contributions

M.H, V.M.-Y.L, and A.J.D. designed the project. A.J.D. and P.D. expressed and purified the recombinant protein with assistance from N.E.M. H.X. and L.C. extracted and purified the human tissue samples and conducted mouse neuron assays. A.J.D. and P.D. fibrillized the brain-seeded samples and heparin-fibrillized calibration samples, with guidance from H.X. P.D. measured the TEM images. H.X. conducted sedimentation assays. A.J.D. and P.D. conducted and analyzed the solid-state NMR experiments. All authors interpreted the results of the study and wrote the paper.

## Competing interests

The authors declare no competing interests.

## Ethics approval

All animal protocols were approved by the University of Pennsylvania's Institutional Animal Care and Use Committee (IACUC).
