## [Peer Review File · Nature Communications]

REVIEWER COMMENTS

Reviewer #1 (Remarks to the Author):

The manuscript by Dregni et. al. focused on tackle the problem on how 4R and 3R in AD tau tangles were mixed. It is a very difficult task. The authors first extracted the AD tau tangles from AD patients as the seeds, then the fibril were amplified and solid-state NMR REDOR was applied. This is a novel application using REDOR to obtain the information on 4R/3R mixing. The statistical explanation was written clearly. I think it is a very good paper.

I only have some minor suggestions.

1. I think if the REDOR technique applicable here, it will also have other pre-requirements. Since ^{13}C - ^{15}N polarization transfer between the intermolecular hydrogen bonds is the key in this REDOR application, we have to assume the fibril core structure is similar between 4R and 3R in the number of residues involved in the intermolecular hydrogen bonding. The hydrogen bonding pattern would not change in the intermolecular interactions between different type of molecules once 4R and 3R are mixed. So no matter 4RN-3RC, or 3RN-4RC or the pure type, the number of these intermolecular hydrogen bonding is not changed. It may be the case here. The author did not explain much on the difference between 3R and 4R in the structure. I think since there were many reports on different types of tau fibril structures, the author cannot assume we know 3R and 4R so well. More information on the structures on 3R and 4R are needed. The brain derived tau structure AD PHF fibril (PDB: 5O3L) was used in this paper. I am not sure the structure in this paper would be the same as this published structure.

2. 10 patients' brain extract were pooled together and used. Since there was difference on these patients' onset ages and disease duration, and clinical diagnosis, I am not sure whether these extracts from different patient would represent the same type of tau fibril structure or not. However, their seeding structure seemed repeatable, suggesting the seeds only produce one type of structure. However, it is not clear for me whether the structure would be similar to the published 5O3L structure.

3. What are those circle structures in TEM images (extended data fig 2, [AD]3R and seeds after 7 days)

4. PHF1 antibody showed a distribution of bands with different sizes showing in the SDS-PAGE gel. Could you please give some explanation on these?

Reviewer #2 (Remarks to the Author):

The paper by Dregni et al represents a sound and significant contribution to the understanding of fibril formation relating to dementia. Supported by extensive biophysical and EM characterization, they present solid-state NMR data which shed new detailed light on the organization of monomers in fibrils with particular focus on how 4R and 3R tau proteins mixes in the fibril upon seeding with AD brain-derived tau – differing significantly from fibrils assembled using heparin.

Overall, the experiments and conclusions seem sound, and the outcome important for the community. As such I recommend the paper being accepted for publication in Nature Communications upon consideration of

(1) I recommend authors more carefully address the significant discrepancy between experimental and simulated REDOR curves as given in Extended Data Fig. 3. I do agree that comparison with the experimental curves obtained for samples with different labelling ratios may provide a "safe" ground for the conclusion – but still it is puzzling that the simulations do not better match the experimental data. The dephasing in the experimental curves seems to be significantly faster both in the initial and later parts of the dephasing period – which may not be fully accounted for by the explanation of non-included long-range effects. Could it be associated with non-perfect decoupling or other? On the experimental side, can the comparison with experimental data for differently labelled ON4R

heparin-fibrillized tau samples be non-adequate for AD-brain derived mixed 4R and 3R samples? As all curves shift significantly (e.g. the experimental 40:60 curve resembles the simulated 60:40 curve), I recommend the authors explore this in more detail (including commenting on effects from fuzzy fibril regions), considering the fundamental importance of the REDOR decay profiles in building the conclusions of the paper.

(2) In light of (1), the authors should discuss the accuracy of the incorporation level percentages. For example, it is argued that a $p_{4 \rightarrow 3}$ value is 0.37 ± 0.03 and k_{hi_4} is $60 \pm 2\%$.

Reviewer #3 (Remarks to the Author):

Recent breakthroughs in cryo-EM technology have allowed the visualization of atomic structures of tau filaments extracted from Alzheimer's disease patient brains; however, how different 3R to 4R isoforms are molecularly mixed in the fibrils remains unknown. In this manuscript, Aurelio and colleagues have utilized solid-state NMR to determine how 3R and 4R tau isoforms could potentially be recruited into the AD fibrils. In addition, the authors found that the ratio of 4R:3R incorporated into AD fibrils is in a proportion of 60:40 with a homotypic preference, and a structural difference exists between heparin-induced tau fibrils and seeded tau fibrils from AD samples.

To characterize how tau isoforms are structurally incorporated into AD seeds, the authors pooled insoluble fractions from the brains of 10 AD patients and then added ¹⁵N or ¹³C labeled recombinant tau monomers into the mixture. Previous studies have shown that tau seeds are heterogeneous, and different tau species could exist between patients even if their Braak staging is similar. If the pooled solution of AD seeds contained different tau fibril strains, how tau monomers are recruited into the seeds may differ depending on the fibril imaged.

The authors' findings of the fluent mixing of 3R and 4R into the AD seed is novel; however, this incorporation pattern may occur differently in vivo. Because the tau monomers the authors used were purified from bacterial cells, they were not post-translationally modified like in human brains. These post-translational modifications, such as phosphorylation, may play an essential role in electrostatic interactions and steric hindrances between the tau monomers and insoluble tau fibrils, which are missing in the author's system. While the use of human samples as seeds may give this study more human relevance, the fact that these measurements are made on in vitro derived fibrils should be more clearly stated in the abstract/conclusions.

Specific Comments:

1. While the introduction provided succinctly describes the need for the study, it is extremely brief.

2. The order of how the result is presented does not follow the order of the figures. The authors defined the mixture of AD fibrils and 4R and 3R tau for Figure 3 after they were used in figure 1 results.

3. In Figure 1.b, why is the [AD] 3RNC western blot not shown in the main figure, but moved to the supplemental data section?

4. In Figure 1.b and extended data figure 1, the authors showed that recombinant tau monomers are incorporated into insoluble mass after 7 days and stained the western blot with total tau antibody and phospho-tau antibody. It would be informative to address why more phosphorylated tau are present (more intense PHF1 bands) after incubating the tau seeds with monomers. Are kinases present in their mixture ?

5. In the primary neuron seeding experiment, that the localization of the AD tau or seeded tau fibrils is not distinguished (for example within the neurons or sitting on top of the neurons). In addition, the images of the neurons are not clear. Consider adding neuronal staining to show localization of tau fibrils and the purity of the culture.

6. Was an equal portion of each brain used to make the pooled insoluble tau sample?

Minor Comments:

1. In the bar graphs of Figure 1D, what are the error bars showing?

2. In Figure 2a: TEM image quality makes it difficult to see. Extended data figure 2 of TEM images seemed to have a better quality images compared to Figure 2.

Reviewer #1 (Remarks to the Author):

The manuscript by Dregni et. al. focused on tackle the problem on how 4R and 3R in AD tau tangles were mixed. It is a very difficult task. The authors first extracted the AD tau tangles from AD patients as the seeds, then the fibril were amplified and solid-state NMR REDOR was applied. This is a novel application using REDOR to obtain the information on 4R/3R mixing. The statistical explanation was written clearly. I think it is a very good paper.

I only have some minor suggestions.

1. I think if the REDOR technique applicable here, it will also have other pre-requirements. Since ^{13}C - ^{15}N polarization transfer between the intermolecular hydrogen bonds is the key in this REDOR application, we have to assume the fibril core structure is similar between 4R and 3R in the number of residues involved in the intermolecular hydrogen bonding. The hydrogen bonding pattern would not change in the intermolecular interactions between different type of molecules once 4R and 3R are mixed. So matter 4RN-3RC, or 3RN-4RC or the pure type, the number of these intermolecular hydrogen bonding is not changed. It may be the case here. The author did not explain much on the difference between 3R and 4R in the structure. I think since there were many reports on different types of tau fibril structures, the author cannot assume we know 3R and 4R so well. More information on the structures on 3R and 4R are needed. The brain derived tau structure AD PHF fibril (PDB: 5O3L) was used in this paper. I am not sure the structure in this paper would be the same as this published structure.

Thanks for this interesting question. We have now added a paragraph in the Discussion to address this question:

" The 4R and 3R tau mixing probabilities that we measured from the REDOR data are based on the assumption that the number of residues that contributes to the 13-sheet core is the same for 4R and 3R tau in the AD-seeded fibril mixture. If the number of residues were different, for example, if 4R tau incorporates more 13-sheet residues into the fibril core than 3R tau, then the $[\text{AD}]4\text{R}^{\text{N}}3\text{R}^{\text{C}}$ REDOR S/S_0 intensities would be biased high, and the true $p_{4\rightarrow 3}$ value would be larger than the measured value. Conversely, $[\text{AD}]3\text{R}^{\text{N}}4\text{R}^{\text{C}}$ REDOR S/S_0 intensities would be biased low, and the true $p_{3\rightarrow 4}$ value would be smaller than the measured value. However, three lines of evidence argue against the different fibril core size between the two isoforms. First, tau filaments from multiple AD brains all show a single 13-sheet core, without any sign of structural heterogeneity {Fitzpatrick, 2017 #3;Falcon, 2018 #82}. Second, the 2D hNH dipolar and INEPT spectra (**Fig. 4**) show similar peak patterns for the two mixed isoform samples, which are also similar to the single isoform samples. Thus, the AD tau seeds are able to propagate by either isoform to develop the same structure. Third, the AD tau seeds are well amplified in our current study, which would be difficult to achieve if a monomer of the opposite isoform is added to the growing fibril with a different core size than the current isoform.

2. 10 patients' brain extract were pooled together and used. Since there was difference on these patients' onset ages and disease duration, and clinical diagnosis, I am not sure whether these extracts from different patient would represent the same type of tau fibril structure or not. However, their seeding structure seemed repeatable, suggesting the seeds only produce one type of structure. However, it is not clear for me whether the structure would be similar to the published 5O3L structure.

We have now added these sentences in the Discussion:

" Because these NMR data were obtained from AD seeds pooled from multiple patient brains, the 60 : 40 mixing and the probabilities we measured reflect the general properties of AD brain tau fibrils instead of being specific to one patient brain. The AD-seeded tau samples show highly reproducible 2D NMR spectra as well as reproducible REDOR dephasing, indicating that there is no detectable molecular structural variation among the multiple AD brains. This is consistent with the recent cryoEM observation that multiple AD brains exhibit the same C-shaped

conformation for the tau fibril core {Falcon, 2018 #82}. These results together support the notion that AD brain tau tangles have a single molecular structure. "

Although we do not attempt to determine the molecular structure of AD seeded tau studied here, the similarity of the twisted morphology of the seeded samples to AD PHF filaments and the reproducibility of AD brain tau structures in multiple brains seen by Goedert and Scheres suggest that the seeded samples likely have the same structure as 5O3L.

0. What are those circle structures in TEM images (extended data fig 2, [AD]3R and seeds after 7 days)

This is a good question. We do not know exactly what these circular objects are, but note that they are ubiquitous in the TEM images of AD brain extracted tau, for example, see an image from the Crowther's 1991 PNAS paper. The objects marked by yellow arrows have an O.D. of ~13 nm and an I.D. of ~5 nm, and are absent in *in vitro* he parin-fibrillized tau samples or tau samples made in the presence of protease inhibitors. Therefore, we suspect that these objects are impurities from the AD brain extraction procedure.

1. PHF1 antibody showed a distribution of bands with different sizes showing in the SDS-PAGE gel. Could you please give some explanation on these?

We assume the reviewer is referring to the immunoblots in Figure 1b and Extended Data Figure 1. We have now added this clarification in the experimental section:

" The 17025 antibody was raised with recombinant human tau, thus it has a higher affinity for recombinant tau than AD brain PHF tau. PHF1 is more specific to the S396/S404 phosphorylated tau, and the bands in PHF1 blots (**Fig. 1b**) mostly represent hyperphosphorylated human tau seeds. As there are six isoforms of human tau high molecular-weight tau species in the AD tau seeds, the banding pattern of PHF1-tau is much more complex than the pattern of recombinant human tau (17025-tau). "

For these reasons, the positions of PHF1 positive bands are expected to differ from the 17025 positive bands.

Reviewer #2 (Remarks to the Author):

The paper by Dregni et al represents a sound and significant contribution to the understanding of fibril formation relating to dementia. Supported by extensive biophysical and EM characterization, they present solid-state NMR data which shed new detailed light on the organization of monomers in fibrils with particular focus on how 4R and 3R tau proteins mix in the fibril upon seeding with AD brain-derived tau – differing significantly from fibrils assembled using heparin.

Overall, the experiments and conclusions seem sound, and the outcome important for the community. As such I recommend the paper being accepted for publication in Nature Communications upon consideration of

We are glad the reviewer appreciates the importance of the work and agrees with the conclusion.

(1) I recommend authors more carefully address the significant discrepancy between experimental and simulated REDOR curves as given in Extended Data Fig. 3. I do agree that comparison with the experimental curves obtained for samples with different labelling ratios may provide a “safe” ground for the conclusion – but still it is puzzling that the simulations do not better match the experimental data. The dephasing in the experimental curves seems to be significantly faster both in the initial and later parts of the dephasing period – which may not be fully accounted for by the explanation of non-included long-range effects. Could it be associated with non-perfect decoupling or other? On the experimental side, can the comparison with experimental data for differently labelled 0N4R heparin-fibrillized tau samples be non-adequate for AD-brain derived mixed 4R and 3R samples? As all curves shift significantly (e.g. the experimental 40:60 curve resembles the simulated 60:40 curve), I recommend the authors explore this in more detail (including commenting on effects from fuzzy fibril regions), considering the fundamental importance of the REDOR decay profiles in building the conclusions of the paper.

We thank the reviewer for this question, which has prompted us to provide additional numerical simulations to explain our reasoning as to why the experimentally measured REDOR curves from the calibration samples represent the gold standard in true REDOR dephasing for given mixing ratios. In short, the numerical simulations deviate from the experiments because no program can capture all the ^{13}C spins near a ^{15}N in a cross- β fibril with realistic computational time while still taking into account multiple spin interactions quantum mechanically. We have now added this additional analysis in Extended Data Figure 3. The key take-home message is that SpinEvolution under-estimates dipolar dephasing whereas second moment analysis over-estimates dipolar dephasing. Below is one paragraph in the Methods section that summarizes these simulations:

“The 9-spin SpinEvolution simulations result in slightly higher S/S_0 values whereas the 48-spin secondary moment analysis result in lower S/S_0 values than the experimental data. The 9-spin second-moment analysis approaches the experimental data the closest. The faster dephasing of the simulated REDOR curves with 48 nearest ^{13}C spins (**Supplementary Fig. 3f**) compared to dephasing by 9 ^{13}C spins indicate that ^{15}N - ^{13}C distances between 5.8 Å and 8.8 Å cannot be neglected. Thus, SpinEvolution simulation for up to 9 ^{13}C spins is an underestimate. On the other hand, the explicit SpinEvolution simulation with 9 ^{13}C spins decays more slowly than the second moment analysis for 9 spins, indicating that other spin interactions, including ^{13}C - ^{13}C dipolar couplings and chemical shifts, moderately slow down the ^{15}N - ^{13}C dipolar dephasing. This means the second-moment analysis of 48 ^{13}C spins overestimates the true decay rate. Therefore, neither simulation approach fully accounts for both long-range ^{13}C - ^{15}N distances and all nuclear spin interactions. Due to these limitations, we rely on the experimentally measured calibration REDOR curves to extract the mixing probabilities of AD-tau seeded filaments accurately.”

We now show the ^{13}C natural abundance corrected experimental data in Figure S3:

“Among the measured REDOR calibration curves from the heparin-fibrillized 0N4R tau samples, the curve of the 100% ^{15}N -labeled 0N4R tau without ^{13}C -labeled protein already exhibits dipolar dephasing to 0.84 at 25 ms (**Fig. 3e** and **Supplementary Fig. 3a**). This indicates that natural abundance ^{13}C spins cause non-negligible dephasing

to ^{15}N . This natural abundance effect can be approximately corrected by dividing all measured REDOR intensities by the intensities of the 100% ^{15}N -labeled sample (**Supplementary Fig. 3b**)."

As to the validity of using calibration samples of heparin-fibrillized 0N4R to calibrate the AD seeded samples' REDOR data, we first note that our hNH-REDOR experiment is conducted with a short 400 μs ^{15}N - ^1H CP step, which should be sufficient to make the contributions of the fuzzy domains to the REDOR negligible. Under this condition, we are primarily observing the dephasing within the parallel-in-register 4.8 Å cross- β amyloid core. Whatever the size and shape of the cores in heparin fibrillized 0N4R tau and [AD]3R/4R tau, the REDOR dephasing is primarily reporting the rigid core contacts from amide nitrogens to the carbons of the same residue or flanking residues in the monomers above or below the ^{15}N -labeled chain in the fibril. Thus, heparin-fibrillized 0N4R tau, where all the monomers are chemically identical, represents an excellent calibration sample for the REDOR experiment. Even if one could make heparin-fibrillized tau using 0N4R and 0N3R monomers with different ratios, the uncertainty of the incorporation ratio would make it impossible to obtain a better calibration curve than what we currently have.

(2) In light of (1), the authors should discuss the accuracy of the incorporation level percentages. For example, it is argued that a $p_{4 \rightarrow 3}$ value is 0.37 +/- 0.03 and k_{hi_4} is 60 +/- 2%.

As explained in (1), we believe the experimental REDOR calibration by 0N4R tau is the best approach.

Reviewer #3 (Remarks to the Author):

Recent breakthroughs in cryo-EM technology have allowed the visualization of atomic structures of tau filaments extracted from Alzheimer's disease patient brains; however, how different 3R to 4R isoforms are molecularly mixed in the fibrils remains unknown. In this manuscript, Aurelio and colleagues have utilized solid-state NMR to determine how 3R and 4R tau isoforms could potentially be recruited into the AD fibrils. In addition, the authors found that the ratio of 4R:3R incorporated into AD fibrils is in a proportion of 60:40 with a homotypic preference, and a structural difference exists between heparin-induced tau fibrils and seeded tau fibrils from AD samples.

To characterize how tau isoforms are structurally incorporated into AD seeds, the authors pooled insoluble fractions from the brains of 10 AD patients and then added ^{15}N or ^{13}C labeled recombinant tau monomers into the mixture. Previous studies have shown that tau seeds are heterogeneous, and different tau species could exist between patients even if their Braak staging is similar. If the pooled solution of AD seeds contained different tau fibril strains, how tau monomers are recruited into the seeds may differ depending on the fibril imaged.

This is an interesting question. We have now added a paragraph in the first paragraph of the Discussion section to address this question.

" Because the NMR data were obtained from AD tau seeds pooled from multiple patient brains, the 60 : 40 mixing and the probabilities we measured reflect the general properties of AD brain tau fibrils instead of the property of a single patient brain. The AD-seeded tau fibrils show highly reproducible 2D NMR spectra as well as reproducible REDOR dephasing, suggesting that there is no detectable molecular structural variation among the multiple AD brains. Although some biochemical studies have reported certain levels of heterogeneity in AD brain tau, these observations were all made based on total tau, including both pathogenic and non-pathogenic tau, in AD patients^{32,33}. Our study instead focuses on sarkosyl-insoluble tau, which has been shown to have highly consistent fibril core structures across multiple AD patient cases⁵, and which exhibits consistent seeding abilities *in vitro* and *in vivo*¹⁸. Therefore, these biochemical and biophysical data provide additional evidence for the isoform-independent molecular structure observed here. "

The authors' findings of the fluent mixing of 3R and 4R into the AD seed is novel; however, this incorporation pattern may occur differently in vivo. Because the tau monomers the authors used were purified from bacterial cells, they were not post-translationally modified like in human brains. These post-translational modifications, such as phosphorylation, may play an essential role in electrostatic interactions and steric hindrances between the tau monomers and insoluble tau fibrils, which are missing in the author's system. While the use of human samples as seeds may give this study more human relevance, the fact that these measurements are made on in vitro derived fibrils should be more clearly stated in the abstract/conclusions.

We agree that our data obtained on bacterially produced recombinant tau do not probe whether and how posttranslational modifications affect the 4R/3R tau mixing in human brains. On a molecular level, it is difficult to imagine how PTMs would influence the mixing of 3R and 4R tau in AD brains. This question would require future investigations.

Specific Comments:

1. While the introduction provided succinctly describes the need for the study, it is extremely brief.

The original text was written for the journal Nature, which prizes brevity and puts background information into the abstract. We have now shortened the abstract and expanded the introduction according to the Nature Communications format.

2. The order of how the result is presented does not follow the order of the figures. The authors defined the mixture of AD fibrils and 4R and 3R tau for Figure 3 after they were used in figure 1 results.

We have updated the results and the captions for figure 1.

3. In Figure 1.b, why is the [AD] 3RNC western blot not shown in the main figure, but moved to the supplemental data section?

This is due to space limitation. Since Extended Data Figure 1 shows all four samples' western blot, we do not feel that the main text figure should enumerate all details. All four main text figures are already very large, so we try to delegate the less important details to extended data figures.

4. In Figure 1.b and extended data figure 1, the authors showed that recombinant tau monomers are incorporated into insoluble mass after 7 days and stained the western blot with total tau antibody and phospho-tau antibody. It would be informative to address why more phosphorylated tau are present (more intense PHF1 bands) after incubating the tau seeds with monomers. Are kinases present in their mixture ?

To demonstrate homogeneity in tau filament sizes, the AD-tau seeds (PHF-1 positive bands) were sonicated intensely before being added to the cell-free seeding reactions. Because of this intense sonification step, ultracentrifugation at 100,000 g for 30 min in the sedimentation assay could not completely sediment the insoluble AD-tau seeds at day 0 and a significant amount of tau seeds stay in the supernatant. After 7 days of seeding reaction, tau filaments were elongated and became easier to be pelleted, resulting in a shift in the amount of AD-tau in the pellet fraction. However, the total amount of AD-tau (supernatant + pellet) was not significantly altered before and after the reaction. Thus, there is no evidence of kinases existing in the insoluble tau seeds.

We have now added this clarification in the Methods section on sedimentation assays.

5. In the primary neuron seeding experiment, that the localization of the AD tau or seeded tau fibrils is not distinguished (for example within the neurons or sitting on top of the neurons). In addition, the images of the neurons are not clear. Consider adding neuronal staining to show localization of tau fibrils and the purity of the culture.

In the neuron-based activity assay, we used R2295M mouse tau specific antibody to reveal the seeded mouse tau aggregates in primary mouse neurons. The neurons were treated with 1% HDTA, a detergent, to remove soluble proteins in the cell culture. This paradigm allows us to specifically show the amount of aggregated mouse tau induced by pathogenic tau seeds without the confounding factor of exogenous tau seeds. To demonstrate the presence of the exogenous human tau and endogenous mouse tau, below we attach immunofluorescence images showing neurons treated with AD tau for 14 days and stained with AT8 (S202/T205 pTau) for AD-tau and R2295M for mouse tau. pTau immunoreactivity shows homogenous distribution of AD-tau seeds, while mouse tau antibody only reveals aggregated mouse tau (a). Further evidence could be found in our previous publication {Xu, 2021 #4}. To elucidate the presence of aggregated mouse tau in cultured neurons and their localization, we show below images of neurons treated with AD-tau seeds for 14 days and extracted with methanol to remove immunoreactivity from soluble mouse tau while preserving cytoskeleton proteins, followed by immunofluorescent staining with monoclonal antibody T49 (for mouse tau), MAP2 (in dendrites) and NFL (in axons) (b). These images show that insoluble mouse tau is primarily surrounding MAP2 and NFL in neurons, indicating their neuronal localization.

Characterization of mouse tau pathology in neuron-based activity assay. CD1 mouse primary hippocampal neurons (PHN) were treated with AD-tau for 14 days at DIV7, fixed 1%HDTA + 4% PFA and stained with R2295M anti-mouse tau antibody for mouse tau aggregates and AT8 anti-ptau antibody for AD-tau seeds (a). Scale bar = 25 μ m. PHN treated the same way were fixed with methanol and stained with T49 antibody for mouse tau, NFL antibody for neurofilament light chain, and MAP2 (b). Scale bar = 10 μ m.

6. Was an equal portion of each brain used to make the pooled insoluble tau sample?

Brain lysates were prepared according to the availability of the fresh frozen tissues from each case. Please find this information added in extended data table 1.

Minor Comments:

1. In the bar graphs of Figure 1 D, what are the error bars showing?

We have now clarified the figure caption as:

" Statistical analysis was performed using One-way ANOVA. Multiple comparison between groups was conducted using Tukey post hoc test. On the graph, * stands for $P < 0.05$, ** stands for $P < 0.01$, *** stands for $P < 0.001$ in the multiple comparison tests. Each data point stands for one biological repeat. "

2. In Figure 2a: TEM image quality makes it difficult to see. Extended data figure 2 of TEM images seemed to have a better quality images compared to Figure 2.

We thank the reviewer for this comment. We have now swapped the two images, to place the full TEM views in figure 2 of the main text, and the aligned fibril images in Extended data Figure 2.

REVIEWERS' COMMENTS

Reviewer #2 (Remarks to the Author):

The work of the authors revising the manuscript in view of reviewers comments is appreciated. Overall the paper stands clearer now.

Still, however, remains the dilemma between using heparin-fibrillized ON4R samples or numerical simulations of REDOR spectra to compare with the corresponding spectra from AD-brain derived mixed 4R and 3R samples:

1) Indeed the authors included many more simulation/moment analysis data in an attempt to get at better match to the experimental curves. This is highly appreciated, however, the representation of data in Supplementary Figure 3 and the long discussion on what resembles best the experimental spectra are very difficult to evaluate - not least because the earlier existing direct overlay of experimental and simulated data now is taken out, leaving the reader to compare decaying signals in different panels of supplementary figure 3 when following the discussion. I would recommend the authors helping the reader in evaluating the quality of the simulations by adding a few figure panels overlaying experimental and the chosen simulated data, and on these clearly point out which match is chosen for the conclusions relating to the different experimental data - i.e., the chosen x:y experimental and x:y simulated curves. This may reduce the length of the discussion and certainly give the reader the opportunity to follow the connect between data and simulations.

2) Also when saying that comparison with experimental spectra is chosen as the solution, it seems important to comment on potential challenges comparing heparin-fibrillated samples with AD-seed samples.

Reviewer #3 (Remarks to the Author):

The authors have addressed our comments

REVIEWERS' COMMENTS

Reviewer #2 (Remarks to the Author):

The work of the authors revising the manuscript in view of reviewers comments is appreciated. Overall the paper stands clearer now.

Still, however, remains the dilemma between using heparin-fibrillized 0N4R samples or numerical simulations of REDOR spectra to compare with the corresponding spectra from AD-brain derived mixed 4R and 3R samples:

1) Indeed the authors included many more simulation/moment analysis data in an attempt to get at better match to the experimental curves. This is highly appreciated, however, the representation of data in Supplementary Figure 3 and the long discussion on what resembles best the experimental spectra are very difficult to evaluate - not least because the earlier existing direct overlay of experimental and simulated data now is taken out, leaving the reader to compare decaying signals in different panels of supplementary figure 3 when following the discussion. I would recommend the authors helping the reader in evaluating the quality of the simulations by adding a few figure panels overlaying experimental and the chosen simulated data, and on these clearly point out which match is chosen for the conclusions relating to the different experimental data - i.e., the chosen x:y experimental and x:y simulated curves. This may reduce the length of the discussion and certainly give the reader the opportunity to follow the connect between data and simulations.

We have now added panels (f-h) in Supplementary Fig. 3 to show overlays of the natural abundance corrected experimental REDOR curves for different calibration samples with three different simulations. These overlays more explicitly show that none of the simulation methods – whether SpinEvolution for up to 9 ^{13}C spins or second-moment analysis for 48 ^{13}C spins - accurately capture the experimentally measured REDOR dephasing curves for the different calibration samples. The second-moment analysis for up to 9 ^{13}C spins approach the experimental data the closest. These inaccuracies in the simulations can all be understood. Therefore, the most accurate way to determine the 4R/3R mixing probability in the AD tau fibril is to use the experimentally measured calibration curves of the $^{15}\text{N}/^{13}\text{C}$ mixed labeled samples, rather than using simulations. This is clarified in the Methods section.

2) Also when saying that comparison with experimental spectra is chosen as the solution, it seems important to comment on potential challenges comparing heparin-fibrillated samples with AD-seed samples.

We have now added a sentence in the Results section to clarify the reason for using heparin-fibrillized tau to calibrate the REDOR dephasing of AD-seeded tau:

" The fact that in vitro fibrillized tau and AD-seeded tau do not have the same rigid core structure does not affect the use of intermolecular REDOR for determining the statistics of mixing, because the REDOR experiment only depends on the existence of parallel-in-register cross- β packing, which is true for both heparin-fibrillized tau and AD tau. "

Reviewer #3 (Remarks to the Author):

The authors have addressed our comments.